# Differential nuclear import sets the timing of protein access to the embryonic genome

Thao Nguyen [1,2,3], Eli J. Costa [2], Tim Deibert [4], Jose Reyes [2], Felix C. Keber [2], Miroslav Tomschik [4], Michael Stadlmeier [2], Meera Gupta[1,2,3], Chirag K. Kumar [2], Edward R. Cruz[2,3], Amanda Amodeo [5], Jesse C. Gatlin [4] & Martin Wühr [1,2,3] ✉

The development of a fertilized egg to an embryo requires the proper temporal control of gene expression. During cell differentiation, timing is often controlled via cascades of transcription factors (TFs). However, in early development, transcription is often inactive, and many TF levels stay constant, suggesting that alternative mechanisms govern the observed rapid and ordered onset of gene expression. Here, we find that in early embryonic development access of maternally deposited nuclear proteins to the genome is temporally ordered via importin affinities, thereby timing the expression of downstream targets. We quantify changes in the nuclear proteome during early development and find that nuclear proteins, such as TFs and RNA polymerases, enter the nucleus sequentially. Moreover, we find that the timing of nuclear proteins' access to the genome corresponds to the timing of downstream gene activation. We show that the affinity of proteins to importin is a major determinant in the timing of protein entry into embryonic nuclei. Thus, we propose a mechanism by which embryos encode the timing of gene expression in early development via biochemical affinities. This process could be critical for embryos to organize themselves before deploying the regulatory cascades that control cell identities.

The oocyte is an exceptionally large cell (-1.2 mm diameter in the frog *Xenopus laevis*) and in many species contains a proportionally large nucleus (Fig. 1a left)[1–4]. After fertilization in frogs, the nucleus becomes tiny compared to the egg, and the nucleocytoplasmic volume ratio (NCV-ratio) drops by more than four orders of magnitude (Fig. 1a, b, Supplementary Fig. 1a)[5,6]. Most nuclear proteins are released into the cytoplasm during this drastic change in nuclear volume, but the total amount and composition of canonical nuclear proteins change little (Fig. 1c, Supplementary Fig. 1b, Supplementary Data 1)[7]. During the rapid early cleavage cycles that divide the single egg into thousands of cells, DNA and total nuclear volume increase approximately exponentially (Fig. 1b)[5], and maternally deposited nuclear proteins from the oocyte nucleus are likely reimported into the newly forming nuclei. Whether all nuclear proteins from the oocyte re-enter the embryonic nuclei simultaneously or if there is a sequential order for nuclear import is unclear. What is known is that once the NCV-ratio reaches a critical value, embryos initiate several essential cellular activities, including the onset of zygotic transcription and cell movement[8–17]. This zygotic genome activation (ZGA) is an ordered process starting with transcripts dependent on Pol III followed by a specific temporal sequence of Pol II-dependent transcripts[13,18–23]. At later stages, the sequenced onset of transcription is often controlled

---

[1]Department of Chemical and Biological Engineering, Princeton University, Princeton, NJ 08544, USA. [2]Lewis-Sigler Institute for Integrative Genomics, Princeton University, Princeton, NJ 08544, USA. [3]Department of Molecular Biology, Princeton University, Princeton, NJ 08544, USA. [4]Department of Molecular Biology, University of Wyoming, Laramie, WY 82071, USA. [5]Department of Biological Sciences, Dartmouth College, Hanover, NH 03755, USA. ✉e-mail: wuhr@princeton.edu

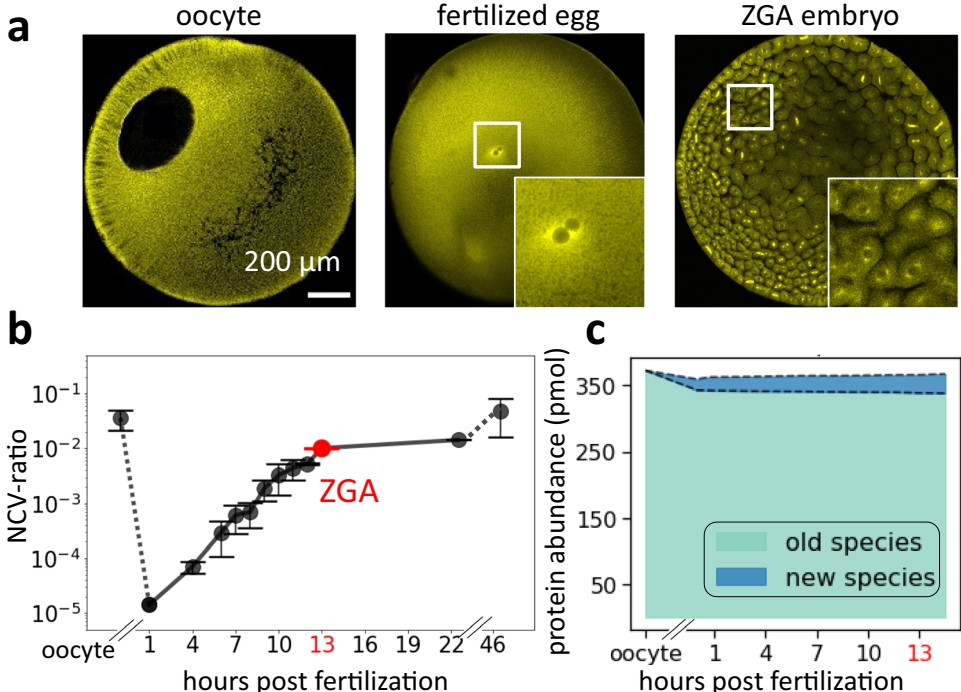

**Fig. 1 | In early embryonic development, nuclear morphology changes drastically while nuclear proteome composition changes little.**
**a** Immunofluorescence images (anti-tubulin) of early frog development show drastic changes in nuclear morphology. The large oocyte (~1.2 mm diameter) contains a proportionally large nucleus (~400 µm). After fertilization, the nucleus is only ~30 µm in diameter. Around zygotic genome activation (ZGA), the embryo contains ~4000 cells with ~18 µm diameter nuclei. The represented images were independently repeated multiple times and observed with similar results. **b** The nucleocytoplasmic volume ratio (NCV-ratio) in early frog development, quantified based on micrographs. The NCV-ratio drops ~10,000-fold from oocyte to fertilized

embryo before increasing exponentially during early cleavage stages. After ZGA, the NCV-ratio gradually increases, approaching the value observed in the oocyte. Solid dots are the mean NCV ratios. Error bars indicate standard deviation. Volume quantification and sample sizes for each time point are provided in Source Data. Embryos developed at 16 °C. **c** Quantification of the fraction of the nuclear proteome replaced by new protein species from oocyte to ZGA. Despite the drastically increasing NCV-ratio from the fertilized egg to ZGA, only ~3% of the nuclear proteome is replaced by newly synthesized canonical nuclear proteins (the main contributors are high abundant histones, which increase ~2-fold).

via a sequenced expression of transcription factors (TFs)[24–26]. However, the inherent time delay between the transcription and translation of new genes is too large to explain the rapid and reproducible onset of different transcriptional events in the earliest developmental stages. Furthermore, many TFs are often maternally deposited and show approximately constant expression levels (Supplementary Fig. 1c, Supplementary Data 1)[7,24]. Thus, it is likely that mechanisms beyond TF cascades are responsible for the sequential onset of gene expression in early development. We postulated that early embryos leverage their changing nuclear morphology by ordering the reimport of nuclear proteins such as TFs to time the onset of rapid downstream events. In this work, we show that in early development maternally deposited proteins' access to nuclei is temporally ordered and that the timing of nuclear import correlates with the onset of a protein's nuclear function. Furthermore, we find that affinity to importins is a major determinant to the ordering of a protein's nuclear entry.

## Results

### Proteins sequentially enter embryonic nuclei
To obtain insight into the times at which proteins gain access to the genome, we measured changes in the nuclear proteome over developmental progression in *Xenopus laevis* embryos. Using a protocol involving rapid filtration that allowed separating nuclei from other compartments, particularly mitochondria (Supplementary Fig. 2), we enriched the nuclei at various embryonic stages and determined the nuclear fraction of each protein (NF) by quantifying its relative abundance in the supernatant and the flow-through via multiplexed proteomics (Fig. 2a)[27,28]. We estimated when half of the protein had entered the embryonic nuclei using a sigmoidal fit of

the protein's NF over time ($T_{\text{embryo}1/2}$). We repeated the experiment for five biological, independent replicates, with a total of 18 time-points and observed overall good agreement between measurements (Supplementary Fig. 3). Figure 2b shows the NF data with sigmoidal fits for three example proteins (Nupl2, Smad2, Polr2b) with $T_{\text{embryo}1/2}$ ranging from 12 h to 31 h, a representative spread for nuclear proteins (Fig. 2c) (we define the time of fertilization as $t = 0$ h). Comparing $T_{\text{embryo}1/2}$ across ~2k canonical nuclear proteins, we found that the timing of nuclear import varied widely (Supplementary Data 2, Fig. 2c). We observed several examples of proteins that were not nuclear in the oocyte[29] but were imported into the embryonic nuclei. Among those, were the origin of the replication complex (Supplementary Fig. 4a), the nuclear pore complex (Supplementary Fig. 4b), β-catenin (Ctnnb1), and the CPC complex (Supplementary Fig. 4c).

Validating the functional significance of our measurements, we found that the subunits of a given protein complex typically enter the nucleus at similar times (Supplementary Fig. 4d). For example, although three different DNA repair complexes (the Fanconi anaemia complex ($T_{\text{embryo}1/2} = 12$ h), the homologous recombination ($T_{\text{embryo}1/2} = 20$ h), and the nonhomogeneous DNA end-joining repair complex ($T_{\text{embryo}1/2} = 35$ h)) enter nuclei at different times, the subunits within a single repair complex arrive together (Fig. 2d). Our observation of delayed sequential nuclear entry suggests that separating repair enzymes from DNA might contribute to the previously observed suppression of DNA repair during the rapid early cleavage cycles[30–32]. Specifically, Hagmann et al. suggested that the DNA-end joining (NHEJ) is dominant in the fertilized egg. However, with increasing amounts of DNA, due

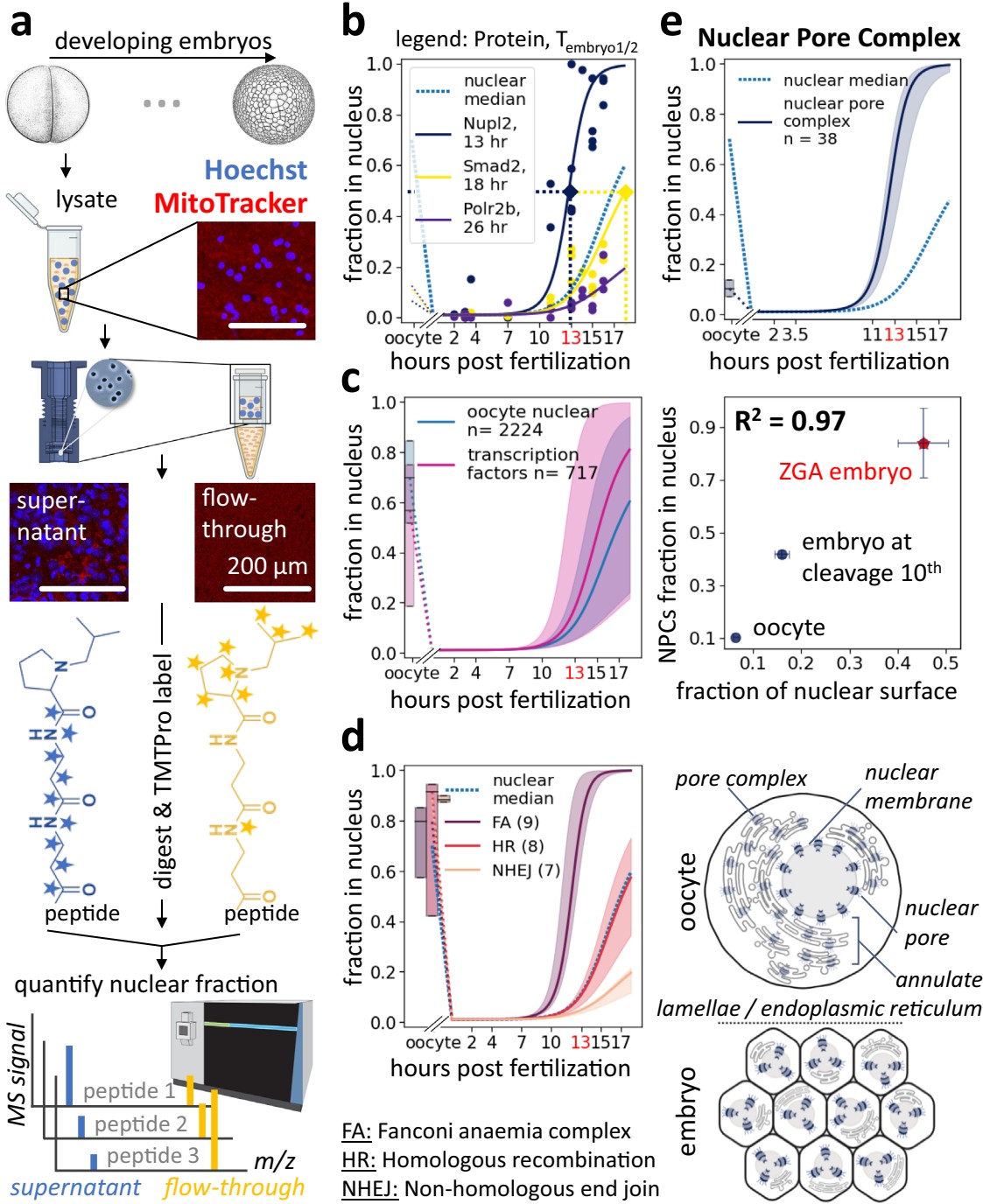

**Fig. 2 | Quantification of nucleocytoplasmic partitioning during early development. a** Assay to quantify nucleocytoplasmic partitioning via multiplexed proteomics. Embryonic lysate, developed at 16 °C, was collected at various stages: nuclei were enriched by filtration through membranes of defined pore sizes. Shown are example epifluorescence images of lysate, supernatant, and flow-through stained with Hoechst (blue) and MitoTracker (red). Supernatant and flow-through fractions were digested into tryptic peptides, labeled with isobaric tags, and subjected to accurate multiplexed proteomic analysis. *Xenopus* illustrations © Natalya Zahn (2022)[86,95]. **b** Nucleocytoplasmic partitioning quantification for three examples. Dots indicate measured fraction of each protein in nuclei. Solid lines indicate fit sigmoids. Blue dashed line indicates median fit from all nuclear proteins. **c** Quantification of time-dependent nuclear fraction for ~2k nuclear proteins and ~600 transcription factors (TFs)[20,71,96]. Solid line indicate the median and shaded area 50% spread. **d, e** Measurements for multisubunit protein complexes. Solid line indicates the median and shaded area 50% spread. **d** DNA repair complexes enter embryonic nuclei at various times, yet their subunits enter simultaneously. Shown

are nuclear entry quantifications for homologous recombination (HR), the non-homologous DNA end-joining repair (NHEJ), and the Fanconi anaemia (FA). An apparent time delay of nuclear entry for some DNA repair complexes explains previous observations that early embryos bypass DNA repair to accommodate fast cell divisions[30–32]. Oocyte nuclear fractions are shown as standard box plots. **e** Change in nucleocytoplasmic partitioning for nuclear pore complex (NPC) subunits. Top: Our nuclear fraction measurement for NPC proteins shows rapid incorporation onto embryo's exponentially increasing nuclear surface. Middle: Quantification of NPC nuclear fraction via MS agrees well with immunofluorescence-quantified nuclear surface change. The *x*-axis is the relative nuclear surface measurement to that of embryos at 22.5 h post-fertilization. Bars indicate standard errors. Bottom: Illustration of dynamic localization of NPC proteins in the oocyte versus the embryo. Oocyte stores NPC subunits in endoplasmic reticulum membranes embedded with pore complexes, called annulate lamellae[34,35].

to cell divisions, homologous recombination (HR) becomes more prevalent. Nonetheless, we characterized a few contrary cases in which individual subunits within a complex enter the nuclei at different times. For example, we found that the essential core subunits of the origin of replication complex (Orc1–5) uniformly enter the embryonic nuclei early ($T_{embryo1/2} = 12$ h), whereas a non-essential component (Orc6) enters at a much later time ($T_{embryo1/2} = 21$ h) (Supplementary Fig. 4a)[33].

Our protocol allows following proteins associated with the nuclear surface and proteins resident in the nucleus itself. We find that nuclear pore complex proteins essential for entry of nuclear resident proteins retain a remarkably constant relationship to the nuclear surface area. In oocytes, nuclear pore complex (NPC) proteins localize predominantly in the cytoplasm (Fig. 2e, Supplementary Fig. 4b). In fertilized embryos, the NPC proteins rapidly incorporate into the exponentially increasing nuclear surfaces. The nuclear accumulation of NPC subunits follows the corresponding changes in nuclear surface area in early development ($R^2 = 0.97$). Specifically, we measured a 7.1-fold increase in the nuclear surface from the oocyte to embryo at the stage of ZGA, corresponding to an 8.1-fold increase in the nuclear accumulation of NPC proteins (Fig. 2e). These findings are consistent with previous electron microscopy studies showing that extra nucleoporins are stored in annulate lamellae in the oocyte

cytoplasm before being incorporated in the embryo's increasing nuclear surfaces (Fig. 2e bottom)[34–36]. These results suggest that import capacity per nuclear surface remains approximately constant during the cleavage divisions, despite the changing pattern of nuclear protein imported.

## Ordered entry of nuclear proteins may explain the timing of downstream activity

Our proteome-wide investigation of the changes in the nuclear proteome during early development revealed that the composition of the embryonic nucleus is dynamic. The ordered access of proteins to the genome could affect the timing of their nuclear functions. We next explored how differential nuclear import might relate to the control of RNA transcription. In frogs and many other species, transcription is inhibited during the early cleavages and is initiated only when the exponentially increasing DNA has titrated out a transcription inhibiting factor from the cytoplasm[8,13,14]. DNA replication factors and histones have been proposed to be the titrated molecules in this model[15,16]. Indeed, we found these proteins were among the earliest proteins to enter the nuclei (Fig. 3a). These factors become depleted from the cytoplasm around the ZGA, such that their concentrations begin to decline in the exponentially increasing nuclei. It has been proposed that the lower concentration of DNA replication factors slows down

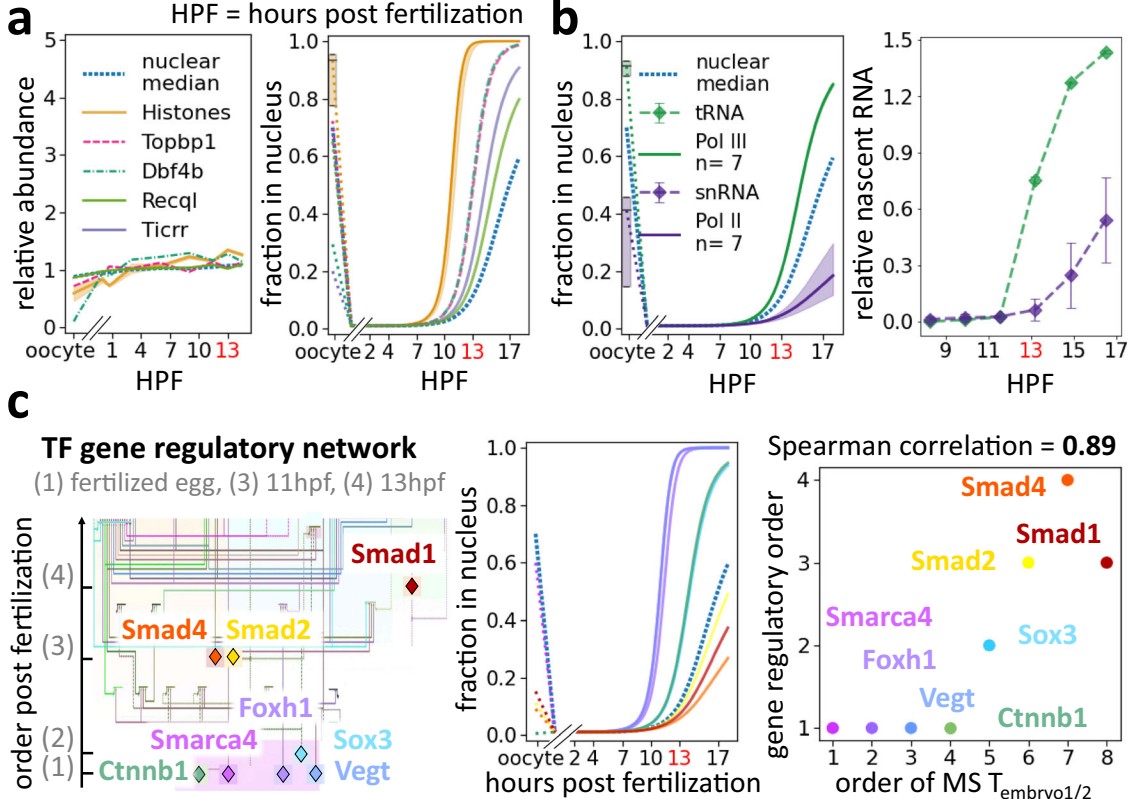

**Fig. 3 | In early development, proteins enter nuclei sequentially correlating with the onset of their nuclear functions. a–c** Embryos developed at 16 °C. **a** Left: Protein abundance dynamics of previously reported ZGA key regulatory factors change little from fertilization to the ZGA[15,16]. Right: These regulators are among the earliest proteins to titrate into embryonic nuclei. Shown are fits of individual ZGA regulators and the median fit of the core histones (H2a, H2b, H3, and H4) and their isoforms with 50% spread. **b** RNA polymerase III (Pol III) and II (Pol II) enter nuclei at different times of development, which corresponds to the respective appearances of their first downstream transcripts. Left: Proteomics data show that Pol III subunits titrate into nuclei before Pol II subunits. Shown are median and 50% spread of unique subunits. Right: The first tRNA transcripts, transcribed by Pol III, and snRNA,

transcribed by Pol II, show corresponding timing to the nuclear entry of their corresponding polymerases. The RNA measurements are quantified from Newport and Kirschner's RNA gel of newly synthesized transcripts[13]. Error bars indicate standard deviation of five snRNA species (snRNA U1, U2, U4, U5, U6). **c** Timing of the nuclear import of TFs in the mesendoderm gene regulatory network corresponds to the timing of their activation. Left: Transcriptional activation order in early *Xenopus* embryos (Adapted drawing from Charney et al.[24] and[43,44]). Middle: MS-quantification of nuclear import in early development for these TFs. Right: Scatter plot of rank between the measured nuclear entry $T_{embryo1/2}$ by MS and the reported temporal gene regulatory network shows strong agreement (Spearman correlation of 0.87, two-tailed *p*-value = 0.005).

DNA replication, thereby enabling the onset of the very first transcripts[16]. Lowering nuclear histone concentrations is believed to change chromatin state, allowing transcription to start[15,37–41].

Even though the embryo can transcribe at the ZGA, only Pol III transcripts are initially observable, followed by a clearly ordered sequence of Pol II transcripts[13,21,22,24]. We investigated whether sequential nuclear import could explain this ordering. We found that while the relative protein abundance of Pol II and Pol III subunits stays approximately constant, Pol III subunits enter embryonic nuclei comparatively early ($T_{embryo1/2} = 15\,h$), and Pol II subunits are delayed ($T_{embryo1/2} = 30\,h$) (Supplementary Data 1,2, Fig. 3b). This timing agrees with the well-established observation that Pol III transcripts (tRNA) are the very first to be transcribed in early development, followed by Pol II transcripts (snRNA and mRNA) (Fig. 3b)[13]. Among Pol III transcripts, we observed a further correlation between the entry of TFs and the transcription of their downstream targets (Supplementary Fig. 5). Gtf3c1–5 are direct transcription factors of tRNA, while Gtf3a is of 5 S rRNA and 7 S rRNA. Gtf3c1–5 enter the nucleus earlier than Gtf3a, which results in the appearance of tRNA before 5 S rRNA and 7 S rRNA[13].

Subsequently, we investigated how the nuclear entry of specific maternally deposited Pol II TFs related to their downstream functions. Charney et al. (2017) described a gene regulatory network for maternal TFs active from fertilization through early gastrulation (Fig. 3c)[24]. Besides the factors mentioned in the diagram, we added Sox3 with the reported rank of Sox7. These proteins were reported to start acting simultaneously and compete for early regulation of Xnr5 gene and Nodal signaling[24,42]. Additionally, we added Smarca4 at the same rank as $\beta$-catenin. Both have been shown to act together to initiate transcription in early embryogenesis[43,44]. We found that these collected TFs do not change their total expression levels in early development (Supplementary Fig. 1c), yet enter embryonic nuclei at vastly different times. Indeed, the order of their nuclear entry is highly predictive of the timing for their downstream gene activities with a Spearman correlation of 0.87 ($p$-value = 0.005) between our measurements of TFs entry into the nucleus and the order in which they activate transcription in embryogenesis (Fig. 3c right). Thus, our analysis can explain why Pol III transcripts precede Pol II transcripts and why individual Pol II transcripts arise in sequence despite their constant expression levels.

## Importin affinities correlate with the timing of nuclear import

We investigated the molecular mechanisms responsible for the differential timing of protein nuclear entry. Proteins could accumulate in the nucleus via active transport through binding to importins, proteins responsible for nuclear import, or by sequestration into the nucleus, e.g., by binding to DNA[6,45]. Differential protein affinities to either DNA or importins might result in ordered nuclear import. However, we found that plasmid DNA affinity was poorly predictive of nuclear entry time (Supplementary Fig. 6a, b).

This prompted us to investigate the role of nuclear transport via importins. Though the early embryo is known to express at least ten different importins, we focused on the most abundant, importin $\beta$ (Kpnb1), and its canonical adapter, importin α1 (Kpna2)[29,46] (Supplementary Fig. 7). We aimed to quantify the proteome-wide binding affinities of proteins to importins by exposing frog egg lysate to varying amounts of active importin $\beta$, controlled by different amounts of added RanQ69L. This constitutively active mutant mimics the GTP-bound form[47]. Ran-GTP causes conformational changes in importin $\beta$ that induces the release of its substrates (Fig. 4a)[48]. Following the addition of importin α1 and one-hour incubation in frog egg lysate, we then isolated importin $\beta$ via affinity purification. Using quantitative proteomics, we measured the abundance of co-isolated proteins and their sensitivity to increasing amounts of RanQ69L (Fig. 4a). We integrated the information of our triplicate measurements by projecting the values on a single dimension, which we call the importin affinity

proxy, determined by cross-validated canonical correlation analysis (Fig. 4b, Supplementary Data 3)[49]. We validated the assay using a set of nuclear localization signal (NLS) peptides with previously measured dissociation constants $K_D$[50] (Supplementary Fig. 8). When we plot the $T_{embryo1/2}$ against the importin affinity proxy, we observed an $R^2$ of 0.46 ($p$-value = 2.6e-22) for NLS-containing proteins (Fig. 4c) or an $R^2$ of 0.29 ($p$-value = 1.7e-46) for the entire proteome (Supplementary Fig. 6c, d)[51]. These $R^2$ values are likely an underestimation of the actual contribution as they do not consider experimental noise in the embryo proteomics or the pulldown experiments[52]. By necessity, our model ignores various layers of nuclear import regulation, which are most likely crucial for embryonic development, including substrate interactions with the other importins, exportins, and importin αs[53,54], changes of expression levels of all proteins involved, and post-translational modifications like phosphorylation of substrates or importins and exportins[55,56]. Nevertheless, our ability to explain at least 46% of the observed variance for the timing of nuclear entry for NLS-containing proteins from this simple assay is remarkable, especially considering that nuclear import in early embryos is undoubtedly more complicated than implied with this analysis.

## A simple model explains the timing of access to the genome in early embryos

Collectively, our data indicate that nuclear proteins enter nuclei at different stages of early development and that the timing of this entry correlates with affinity to importin. Therefore, we postulate that according to competitive binding the limited amount of importin in cells (at ~1.5 μM, compared to ~320 μM of all protein with a predicted nuclear localization signal) binds predominantly to the highest affinity substrates (Fig. 4d, Supplementary Data 4)[51,57]. These substrates are then preferentially imported into early nuclei. With developmental progression, total nuclear volume increases due to nuclear import and an increasing number of nuclei. After high-affinity substrates are depleted from the cytoplasm and imported into the increasing nuclear volume, importins become available to lower affinity substrates, and those proteins can then enter the nuclei. To formalize this hypothesis, we developed a simple model for competitive binding of substrates with varying $K_D$ to a limiting amount of importin. Furthermore, we assume that nuclear import flux is partitioned among substrates based on their relative binding to importin. We derive the net nuclear import flux from immunofluorescence images (Supplementary Fig. 9a).

This straightforward model recapitulates the observed differential nuclear import of nuclear proteins well. It provides a quantitative framework for how the embryo could use substrate affinities to importin to determine the timing of protein's import into the nucleus and, subsequently, their downstream nuclear functions (Fig. 4e). Interestingly, our model predicts that the concentration of nuclear proteins with high affinity to importin is high in the early nuclei and decreases as additional proteins are imported and the nuclear volume increases. Intermediate affinity substrates reach a maximal nuclear concentration at intermediate times before decreasing. Low-affinity substrates are predicted to reach maximal nuclear concentration at late times (Supplementary Fig. 9b). Finally, based on this model, we expect that the sequential nuclear protein import observed during developmental progression should also occur similarly in individual cell cycles after the nuclear envelope reforms.

## Nuclear entry in a single cell cycle following mitosis mimics the sequence in a developing embryo

During mitosis, the nuclear membrane breaks down, and most nuclear proteins dilute into the cytoplasm. If importin affinities govern the global sequence of nuclear entry during the cleavage period, they should also govern re-entry on the shorter time scales following each round of mitosis. To test this possibility, we imaged the nuclear import following mitosis of nine GFP-labeled TFs observed in our embryo

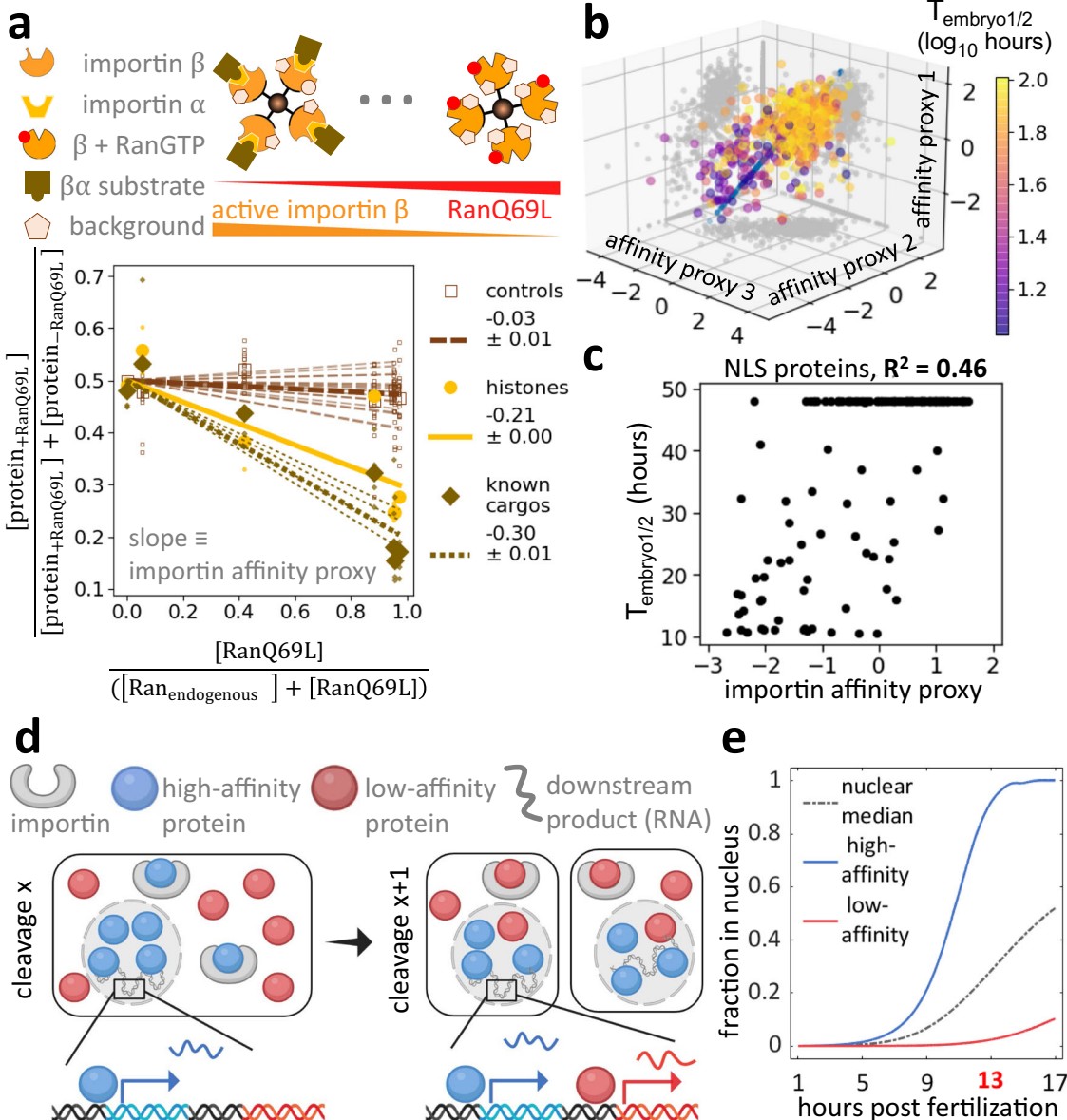

**Fig. 4 | The affinity of proteins to importin contributes significantly to their ordering of nuclear entry in early development. a** Estimation of proteome-wide affinity to importin α/β. We quantified changes in protein abundance associated with importin beads among conditions with varying amounts of RanQ69L. Abundance of known importin α/β substrates[53], including histones, decreases with increasing RanQ69L concentration. Large dots represent the median protein fraction of a protein subgroup at each RanQ69L concentration, while small dots represent measurements for individual proteins. We applied a linear fit for each protein with a fixed *y*-intercept and used the slope to proxy for a protein's affinity to importin. **b** Scatter plot of triplicate affinity proxy measurements from experiments outlined in (**a**). We integrated these measurements to one dimension using cross-validated canonical correlation analysis[49]. **c** Importin affinity can explain a significant fraction of the timing of nuclear entry in early development. The scatter plot shows $T_{embryo1/2}$ versus importin α/β affinity proxy. The observed Pearson correlation suggests that importin affinities can explain >46% of the variance of the timing of nuclear entry or NLS containing proteins in early embryonic development. **d** Schematic of our proposed model in which the differential affinity of proteins to importin controls the timing of genomic access in embryonic development. A high-affinity protein titrates into the nucleus faster than a low-affinity protein, resulting in the corresponding DNA access of proteins. This ordering could determine the timing for the onset of downstream transcription. **e** Simulation of the model proposed in (**d**). We model competitive binding of substrates with varying affinity to a limiting number of importin. The proposed model provides a simple explanation for the timing of protein access to the embryonic genome in early development.

proteomics measurements in cell-free droplets prepared from egg lysates. The droplets were generated using a "T-junction" microfluidic device and many contained demembranated sperm DNA, which spontaneously induces nuclei formation (Fig. 5a left, Supplementary Video 1). Using time-lapse confocal microscopy, we then monitored the nuclear import of GFP-labeled proteins of interest (Fig. 5a right). To facilitate interexperimental comparisons of GFP-protein import rates, we used mCherry tagged with a nuclear localization signal (mCherry-NLS) as an import standard. From the relative nuclear-to-cytoplasmic

intensity measurements, we extracted the half-time ($T_{droplet1/2}$) of nuclear import for each protein and calculated the difference ($\Delta T_{droplet1/2}$) relative to mCherry-NLS observed in the same experiment (Fig. 5b). We observed strong agreement (Spearman correlation of 0.82, *p*-value = 0.007) between the order of nuclear protein import in embryos and the order of nuclear protein entry in cell-free droplets via imaging (Fig. 5c, Supplementary Fig. 9c, d). This agreement holds despite the drastic changes in morphology and the potential differences in protein expression levels and post-translational modifications

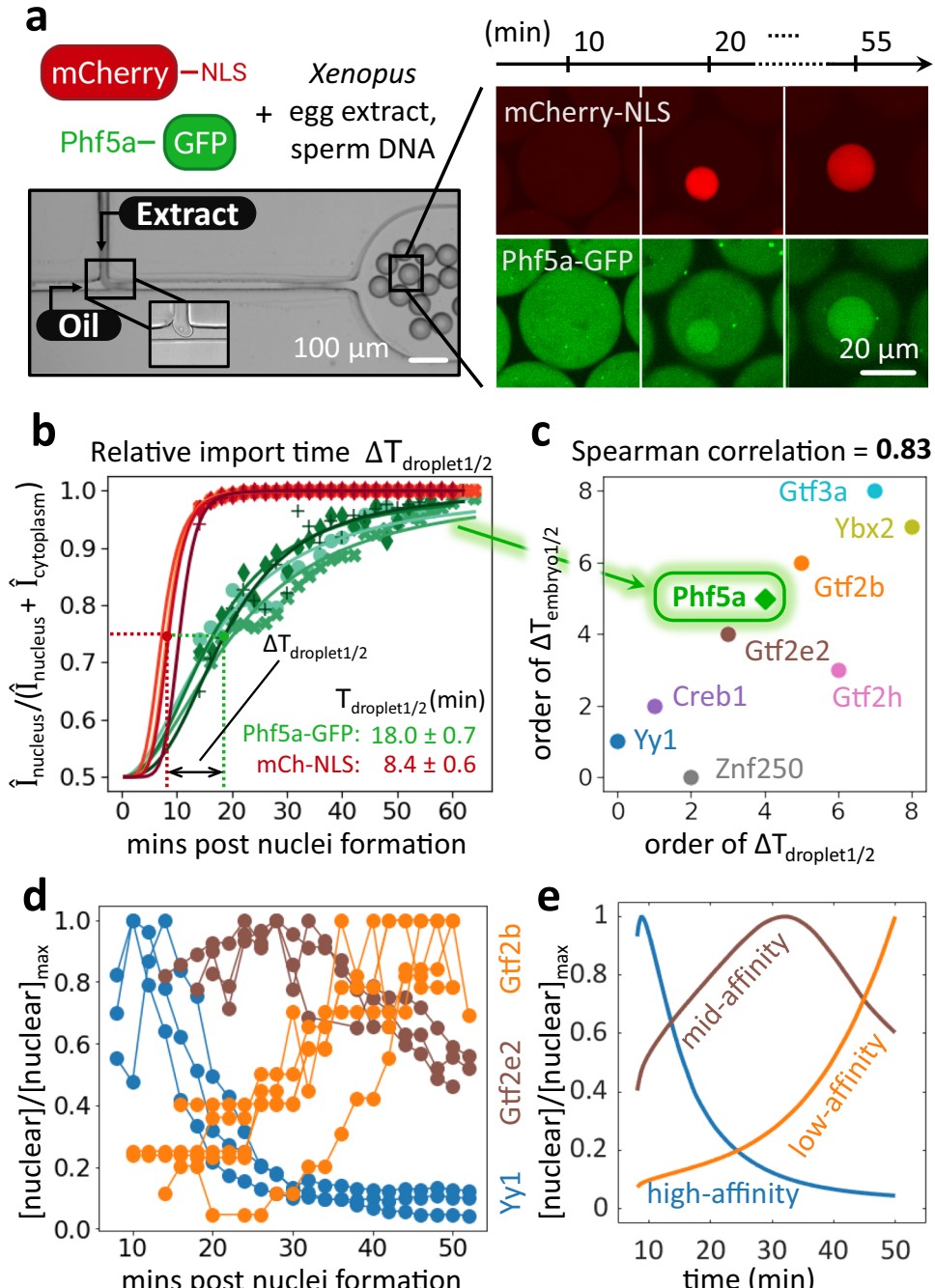

**Fig. 5 | The temporal order of nuclear import in cell-free droplets observed via imaging recapitulates the nuclear entry order observed in the embryo proteomic assay. a** Left: Imaging of nuclear import in cell droplets. *Xenopus* egg extract doped with sperm DNA, which initiates the formation of nuclei, and GFP-tagged protein of interest (here Phf5a*)* and mCherry-NLS were encapsulated in oil droplets with a microfluidic device. Right: We monitored nuclear import kinetics via fluorescence microscopy. **b** We quantified the relative fluorescent signal intensity *I* in the nucleus and cytoplasm and fit the data with a sigmoid to extract the time ($T_{\text{droplet1/2}}$) at which the relative intensity reaches half of its max value. To overcome extract variability, we calculate the import time difference ($\Delta T_{\text{droplet1/2}}$) between mCherry-NLS and the protein of interest. Markers represent the raw measurements. The different symbols represent different droplets; lines are sigmoid fits of corresponding droplets. From these experiments, we extract the median $\Delta T_{\text{droplet1/2}}$. **c** Scatter plot of the order in nuclear import time ($\Delta T_{\text{droplet1/2}}$) from the cell-free assay and the order in $T_{\text{embryo1/2}}$ for the nine TFs show strong agreement (Spearman correlation of 0.82, *p*-value = 0.007). **d** Imaging results of the nuclear protein concentration [nuclear]. The early titrator nuclear protein Yy1 (blue) is high at the early stage and decreases over time, followed by Gtf2e2 (brown) and Gtf2b (orange). **e** In our import model, we predict that nuclear concentration of high-affinity proteins (blue) is high in the early nuclei and decreases with the continuous import of additional nuclear proteins and the increasing nuclear volume. Nuclear proteins with lower affinities (brown then orange) will reach their highest nuclear concentration at some later times and in a sequence corresponding to their interaction strengths to importin. The imaging results are consistent with the model (visualized by colors corresponding to (**d**)).

between the egg and the post-ZGA embryo. Our results suggest that differential nuclear entry could be used as a biological timing mechanism within each cell cycle. The timing of nuclear entry seems to be encoded in the proteins' NLS sequences: for a subset of proteins, we transferred their bioinformatically-predicted NLS sequences[51] to GFP and showed that the timing of nuclear entry of the fluorescent proteins is similar to the original NLS-proteins (Supplementary Fig. 10). Additionally, we find that the nuclear protein concentration of early imported substrates (like Yy1) is initially high and decreases over time (Fig. 5d). Protein Gtf2e2 reaches the maximum concentration at an intermediate stage. Lastly, late protein Gtf2b shows the highest concentration at the end of our measurements. These observations agree well with the predictions of our simple model for proteins with high, intermediate, and low affinity for importin (Fig. 5e).

## Discussion

The development of a fertilized egg to an embryo with a canonical body plan and hundreds of different cell types requires remarkable organization in space and time. While we have learned much about the spatial organization in early embryos, our understanding of embryo organization in time is substantially less refined[24,25,58]. Cascades of TFs provide natural order to the timing of newly expressed genes during cell differentiation[25,59,60]. However, early development occurs at an astonishingly fast speed and is mostly transcriptionally silent. The inherent time delay between the transcription and translation of new genes is not consistent with the rapid and reproducible onset of different transcriptional events. Despite the drastic changes of embryonic morphology in early development, the proteome is remarkably constant (Fig. 1c, Supplementary Fig. 1b)[7]. For example, the TFs responsible for mesendoderm formation show approximately constant expression levels (Supplementary Fig. 1c)[24].

Here, we find evidence that rapid early events in development might be ordered via biochemical affinities of maternally deposited nuclear proteins to importin. We find that proteins believed to inhibit the onset of transcription in early embryonic development partition into the nucleus early (Fig. 3a). Subsequently, they are diluted with increasing nuclear volume. Once the embryo is permissive for transcription, the timing of protein entry into embryonic nuclei correlates strongly with the activation of their nuclear functions (Fig. 3c). We present a model in which nuclear import is partitioned based on relative affinity to importin. This model predicts a similar ordering for nuclear import during a single cell cycle. To test this model, we performed nuclear import assays in droplet encapsulated cytoplasm (Fig. 5). Furthermore, our model predicts that high-affinity proteins decrease their nuclear concentration over time while medium and low-affinity proteins reach maximal nuclear concentrations at later stages (Fig. 5e, Supplementary Fig. 9b). Our model can explain the loss of inhibitory effects on transcription before the ZGA for high importin affinity proteins like histones due to their lowering of nuclear concentrations with increased nuclear volume[15,16]. At the same time, the model can explain how lower affinity transcription factors enter the nuclei only later, controlling the temporal onset of their downstream nuclear functions.

In this study, we quantified the temporal entry of ~2k nuclear proteins. While we could only investigate a limited set of proteins for the timing of their downstream nuclear functions, our results suggest that the embryo widely uses the observed inherent timing mechanism. In addition, over 46% of the observed time-variance in nuclear import across all NLS-containing proteins was explained via differential affinities of the proteins for importin, despite quantifying affinities to only importin $\alpha/\beta$ using a crude and noisy biochemical assay. Regulation of nuclear transport in the embryo must be much more complicated than implied by our simplistic assay and model. Nevertheless, the observed predictive power suggests that this fundamental biochemical mechanism plays a crucial role in the temporal organization of

developing early embryos that could set the stage before gene regulatory networks orchestrate cellular differentiation.

## Methods

### Oocyte, egg, and embryo collection

Mature *Xenopus laevis* females and males were purchased from Nasco and maintained by Laboratory Animal Resources at Princeton University. All animal procedures are approved under IACUC protocol 2070, reviewed in April 2021 by Princeton University Institutional Animal Care and Use Committee. Xenopus females were euthanized for ovary collection in 0.1% aminobenzoic acid ethyl ester (Tricaine, MS222) (Sigma A-5040) and then sacrificed by pitching. For testes collection, an equivalent procedure was followed with male frogs. Females were ovulated with at least 6-month rest intervals. *X. laevis* oocytes, eggs, and testes were collected as previously described[61].

**Oocyte collection.** Female frogs were first primed by injecting 100U of pregnant mare serum gonadotropin (PMSG) into the dorsal lymph sac 3–60 days before the experiment to ovulate eggs. *Xenopus* females were sacrificed as described above on the day of the experiment. Ovaries were collected in a petri dish and cultured in an oocyte culture medium (1 L of OCM: 1 bag of 'Leibovitz's L-15 Medium powder (ThermoFisher Scientific #41300039), 8.3 mL Penicillin/Streptomycin, 0.67 g BSA) with the pH adjusted to 7.7 using NaOH and passed through a 0.22 μm filter[62]. We kept oocytes for up to a week and exchanged OCM daily. Alternatively, in later stages of the study, defolliculated oocytes from Ecocyte Bioscience US LLC were used.

**Egg collection.** At 16 h before egg collection, female frogs were injected with 500 U of human chorionic gonadotropin (HCG) and kept at 16 °C in Marc's modified Ringer's[63] (MMR: 100 mM NaCl, 2 mM KCl, 1 mM MgCl₂, 2 mM CaCl₂, 5 mM HEPES pH 7.8, 0.1 mM EDTA, pH to 7.8 by 10 M NaOH)[63]. We collected eggs the next day in MMR buffer and sorted out pre-activated ones for further use.

**Embryo collection.** For in vitro fertilization, we first isolated testes from male frogs as previously described[61]. The testes were stored in OCM at 4 °C and exchanged daily for up to one week of use. Then, we collected eggs from females onto Petri dishes by gently squeezing the frog. One-quarter of one testis was used per 500 eggs by first crushing and then mixing in the eggs using a sterile pestle. The mixture was incubated at room temperature (RT) for 5 min, followed by another mix and an additional 5 min of incubation. Fertilization was then induced by flooding the eggs with 0.1 × MMR. After about 30 min at 16 °C, embryos uniformly rotated to face their pigmented halves upwards, indicating that fertilization succeeded. After this checkpoint, embryo jelly coats were removed by incubating with 2% L-cysteine in 0.1 × MMR at pH 7.8 with NaOH for 2–5 min or until the jelly coats appeared to be removed. Embryos were then washed thoroughly with 0.1 × MMR to remove any residual cysteine. Embryos develop in 0.1 × MMR at 16 °C until the desired time points.

We staged embryos based on Nieuwkoop and Faber[64].

### Immunofluorescence of oocytes and embryos

The immunofluorescence procedure was performed essentially as previously described[65]. Briefly, at stages of interest, ~20 embryos were arrested at interphase using cycloheximide, then collected and fixed with Methanol/EGTA for 24 h. Embryos were then rehydrated using 25%, 50%, 75%, and 100% TBS (10 mM Tris−HCl, pH 7.4, 155 mM NaCl, and 0.65 g/L of NaN₃ to inhibit bacterial growth) and then in Methanol before bleaching with H₂O₂.

Embryos were re-submerged in TBSNB (TBS + 0.1% Igepal CA-630, 1% BSA, 2% fetal calf serum). Samples were then incubated with α-tubulin (B-5-1-2) (Sigma T6074) that was pre-labeled with Alexa-488 using APEX™ Antibody Labeling Kits (Invitrogen) at a 1:200 dilution in

the dark at 4 °C for 12 h., followed by washes in TBSNB for 24 h and two washes in TBS for 10 min. Finally, samples were dehydrated in Methanol and cleared by Murray's clear (2:1 Benzyl benzoate: Benzyl alcohol) before being mounted on a custom-made mounting slide for confocal imaging analysis[65].

Image analysis was performed on a laser scanning microscopy Zeiss 880 confocal microscope. The acquisition sequences were like previously published protocols[66]. For early-stage embryos, only a single focal plane that captured all cells was acquired. For later-stage embryos with multiple cells, a stacked image at 7.3 μm step size was acquired along the animal and vegetal axis. Images were analyzed using ImageJ-win64 version 1.8.0.

To quantify the nucleocytoplasmic volume ratios, we assumed that embryos are rotationally symmetric and quantified nuclear volume for the embryos' representative sector. We assumed nuclei to be spherical and derived the volume for each nucleus from its measured diameter. We calculated average cell volume by dividing embryo volume by previously reported or newly measured cell numbers[5,64].

## Nuclear filtration

To make embryo extract, embryos at the 2-cell, 4-cell, cleavage 10, 11, 12 (the ZGA), 1 h post-ZGA, and 3 h post-ZGA developmental stages were collected and prepared into lysate mostly as previously described[67]. Briefly, around 200 embryos were collected per time point. Embryos were first arrested in the interphase by incubating in 150 μg/mL cycloheximide for 1 h. and then transferred into a standard 0.2 mL PCR tube filled with ELB (250 mM sucrose, 50 mM KCl, 2.5 mM MgCl2, 10 mM HEPES pH 7.8 with KOH) + 1 μg/mL LPC + 1 μg/mL cytochalasin D + 100 μg/mL cycloheximide. Embryos were packed and crushed at the maximum accessible speed (11,600 g) on a tabletop centrifuge. The cytoplasmic layer was withdrawn with a 23-gauge needle. The embryonic extract was supplemented with 10 μg/mL cytochalasin D, 10 μg/mL LPC, 100 μg/mL cycloheximide, 1 μM nocodazole, energy mix (7.5 mM creatine phosphate, 1 mM ATP pH 7.7, 1 mM MgCl$_2$) at 1:100 dilution. To monitor the quality of the nucleus and label the cytoplasm in the filtration experiment, ~1 μg/mL Hoechst dye, ~0.5 μM NLS-GFP, and 1:1000 dilution of MitoTracker Red (M7512 ThermoFisher) were added to the extract. The extract was stored on ice for further use.

To isolate the nuclei from the cytoplasm, we used 3D filter holders. The design files are available on our GitHub page (https://github.com/wuhrlab/3DFilterHolderDesigns). We used the Hubs platform (https://www.hubs.com/) to print the holders using either Standard Resin (SLA) or Dental resin (SLA) materials at a 20% infill rate, 50 μm layer height.

The undiluted embryonic cell lysate was filtered through polycarbonate membranes with uniform pore sizes (5 μm) using a tabletop centrifuge at 2000 g for 2.5 min at 4 °C. Nuclei remained in the supernatant while nuclear-depleted cytoplasm flowed through the membrane. To further remove cytoplasmic impurities, the supernatant was diluted 2-fold with XB buffer, and the filtration spin was repeated at 2000 g for 2.5 min at 4 °C. The two flow-through fractions were collected and combined. Lysates were incubated with Hoechst, NLS-GFP, and MitoTracker for checking via imaging. At each stage, the nuclear and cytoplasmic fractions were digested into tryptic peptides, labeled with isobaric tags, and subjected to accurate multiplexed proteomics analysis.

Canonical nuclear proteins were defined as proteins that are classified as nuclear in the frog oocyte[4] and various other cell types from published databases such as hyper LOPIT[68], Cell Atlas[69], Protein Atlas[70], and Uniprot[71], To avoid the fraction of proteins that are promiscuously assigned to several subcellular localizations, the list excluded proteins whose assigned localizations also include the following compartments: mitochondria, Golgi, endoplasmic reticulum, cytosol, cytoskeleton, and plasma membrane.

To correct for cytoplasmic proteins retained on the filter, we subtracted the signal of the 2-cell stage supernatant experiment (negligible nuclear amounts) from all other supernatant experiments. The MS analysis of nuclear filtration assays measured the relative protein signal in each fraction at each collected time-point. Using the immunofluorescence (IF) data, we converted the time post fertilization to the nuclear-to-cell volume ratio (NCV-ratio) using the slmengine-function from MathWorks File Exchange, created by John D'Errico. For each protein, the nuclear fraction was fitted by a sigmoidal function of NCV-ratio, assuming the final nuclear fraction value reached the value in the oocyte[29]. The fit parameter (bound by 0 and 1) was defined as the NCV-ratio value when 50% protein amount enters the nucleus. Using the same spline fit of the IF data, this ratio was converted to $T_{embryo1/2}$, defined as the time post fertilization at 16 °C when 50% protein enters the embryonic nuclei.

## Investigation of embryonic nuclear entry times for shared subunits of protein complexes

We employed a similar data analysis approach as previously described by[72]. Briefly, using CORUM core complex databases[73], we identified core complexes of at least five members presented in our dataset of the embryonic nuclear import time series. We calculated the standard deviation of the nuclear entry time of each identified complex. For comparison, we generated a null dataset of random assignment members for each complex. We compared distributions of standard deviations between both data sets using the Wilcoxon-rank test[74].

## Nuclear isolation with a commercial kit

Embryo extract at cleavage 12 (the ZGA) was prepared as described above. The undiluted embryonic extract was projected to the Abcam Nuclear Extraction Kit (ab113474) and followed the manufacturer's suggested protocol. The resulting nuclear and cytoplasmic fractions were digested into tryptic peptides, labeled with isobaric tags, subjected to the MS, and nuclear fraction analysis as described above.

## MS sample preparation and analysis

Samples were prepared mostly as previously described[75]. Lysates were collected in 100 mM HEPES pH 7.2. To reduce disulfides, Dithiothreitol (DTT) (500 mM in water) was added to a final concentration of 5 mM (20 min, 60 °C). Samples were cooled to RT, and cysteines were alkylated by the addition of N-ethyl maleimide (NEM, 1 M in acetonitrile) to a final concentration of 20 mM followed by incubation for 20 min at RT. 10 mM DTT (500 mM stock, water) was added at RT for 10 min to quench any remaining NEM. A methanol-chloroform precipitation was performed for protein clean-up, and the collected protein pellets were allowed to air dry. Samples were taken up in 6 M guanidine chloride in 200 mM EPPS pH 8.5. Subsequently, the samples were diluted to 2 M guanidine chloride in 200 mM EPPS pH 8.5 for overnight digestion with 20 ng/μL Lys-C (Wako) at RT. The samples were further diluted to 0.5 mM guanidine chloride in 200 mM EPPS pH 8.5 and then digested with 20 ng/μL Lys-C and 10 ng/μL trypsin (Promega) at 37 °C overnight.

The digested samples were dried using a vacuum evaporator at RT and taken up in 200 mM EPPS pH 8.0. Then, total material from each condition was labeled with tandem mass tags (as indicated by the experiment: TMT-6plex, TMT-11plex, TMTpro-16plex - Thermo Fisher Scientific). TMT/TMTpro samples were labeled for 2 h at RT. Labeled samples were quenched with 0.5% hydroxylamine solution. Samples from all conditions were combined into one tube, acidified to pH < 2 with phosphoric acid (HPLC grade, Sigma) and cleared by ultracentrifugation at 100,000 × g at 4 °C for 1 hour in polycarbonate tubes (Beckman Coulter, 343775) in a TLA-100 rotor. Supernatants were dried using a vacuum evaporator at RT. For a low complexity sample, dry samples were taken up in HPLC-grade water and stage-tipped for desalting[76] and resuspended in 1% formic acid (FA) to 1 μg/μL for mass spectrometry analysis. For high complexity samples, the supernatant

was sonicated for 10 min and then fractionated by medium pH reverse-phase HPLC (Zorbax 300Extend C18, 4.6 × 250 mm column, Agilent) with 10 mM ammonium bicarbonate, pH 8.0, using 5% acetonitrile for 17 min followed by an acetonitrile gradient from 5% to 30%. Fractions were collected starting at minute 17 with a flow rate of 0.5 mL/min into a 96 well-plate every 38 s. These fractions were pooled into 24 fractions by alternating the wells in the plate[77]. Each fraction was dried and resuspended in 100 μL of HPLC water. Fractions were acidified to pH <2 with HPLC-grade trifluoroacetic acid, and stage-tipping was performed to desalt the samples. For LC-MS analysis, samples were resuspended to 1 μg/μL in 1% FA and HPLC-grade water, and ~1 μg of peptides were analyzed per 1 h run time.

Approximately 1–3 μg of the sample was analyzed by LC-MS. LC-MS experiments were performed with an nLC-1200 HPLC (Thermo Fisher Scientific) coupled to an Orbitrap Fusion Lumos (Thermo Fisher Scientific). For each run, peptides were separated on an Aurora Series emitter column (25 cm × 75 μm ID, 1.6 μm C18) (ionopticks, Australia), held at 60 °C during separation by an in-house built column oven. Separation was achieved by applying a 12% to 35% acetonitrile gradient in 0.125% formic acid and 2% DMSO over 90 min for fractionated samples and 180 min for unfractionated samples at 350 nL/min at 60 °C. Electrospray ionization was enabled by applying a voltage of 2.6 kV through a MicroTee at the inlet of the microcapillary column. As indicated in each proteomics experiment, we used the Orbitrap Fusion Lumos with a TMT-MS3[28], TMTc+[78], TMTpro-MS3[79], or TMTproC[27] as previously described.

Mass spectrometry data analysis was performed essentially as previously described[78]. The mass spectrometry data in the Thermo RAW format was analyzed using the Gygi Lab software platform (GFY Core Version 3.8) licensed through Harvard University. Peptides that matched multiple proteins were assigned to the proteins with the greatest number of unique peptides. TMT-MS3[28], TMTc+[78], TMTpro-MS3[79], or TMTproC[27] data were analyzed as previously described.

### RNA gel analysis
The RNA gel image was extracted from Newport and Kirschner's 1982 publication[13] and analyzed using ImageJ. The relative intensity of each RNA band was measured 5 times and the mean value was reported. For snRNA, there were 5 visibly distinct bands that were each measured individually.

### Importin affinity assay
For importin and Ran constructs, we received plasmids gift for the following constructs: GST-importin α (*Xenopus*) and GST-importin β (*Xenopus*) from Sabina Petry, His-Tev- RanQ69L and ZZ-importin β (*Homo sapiens*) from Dirk Görlich and Thomas Güttler. Proteins were expressed and purified mostly as previously described[80,81]. Briefly, all constructs were transformed into Rosseta2 E. coli cells (Fisher: 71-403-4) for protein expression and were grown in TB media (Sigma: T0918) prepared according to the supplier's instructions. Cells were grown at 37 °C, shaking at 200 RPM. When an OD600 of 0.8 was reached, cells were induced with 0.15 mM isopropyl-β–D-1- thioga-lactopyranoside (IPTG) and grown overnight at 20 °C 200 RPM to reach an OD600 post-induction of 25.6. The culture was harvested the next morning by centrifugation at 4 °C at 5000 g with Beckman J2-MI centrifuge with JA-10 rotor for 20 min. Cells were lysed on ice using 0.25 mg/mL lysozyme in lysis buffer (50 mM K-phosphate pH 7.0, 500 mM NaCl, 5 mM Mg(OAc)₂, 1 mM Ethylenediaminete-traacetic acid (EDTA), 2 mM DTT, 40 U/mL Benzonase Nuclease (Novagen 70746-4) and 2 mM PMSF for 10 min. After lysozyme digestion, the lysate was pipetted up and down until a homogenous mixture was reached. Cells were then further homogenized using an EmulsiFlex (Avestin) in lysis buffer. The lysate was clarified, and the supernatant was collected. Most of the constructs continued to further purification steps, except for ZZ-tag-Importin-β, which was aliquoted in 10–20 μL aliquots, flash-frozen using liquid N₂, and stored at −80 °C for the importin interaction experiment.

For GST-importin α, and GST-importin β constructs, the lysates were bound to Pierce Glutathione Agarose resin (ThermoScientific 16101), washed, and eluted in lysis buffer containing 10 mM Glutathione. The eluents were collected for further purification.

For His-Tev-RanQ69L construct, the lysate was bound to NiNTA agarose beads (Qiagen 1018236), washed in lysis buffer +10 μM Guanosine-5'-Triphosphate Disodium Salt (GTP) (ThermoScientific 56001-37-7), and eluted in lysis buffer containing 200 mM Imidazole +10 μM GTP. The collected eluent of His-Tev-RanQ69L was then subjected to an overnight His-TEV protease (Invitrogen 10127-017) cleavage at 10 U per 100 μg target proteins and dialysis sequence to exchange to final buffer solution of CSF-XB (100 mM KCl, 20 mM HEPES, 2 mM MgCl₂, 0.1 mM CaCl₂, 4 mM Ethylene glycol-bis(2-aminoethylether)-N,N,N',N'-tetra-acetic acid (EGTA), pH7.8) + 30 μM GTP. The dialyzed solution was bound to the 2nd NiNTA column to remove His-TEV protease, and the Ran construct was eluted in the final buffer for further purification.

All protein constructs, after elution, were further purified using gel filtration (Superdex 200 HiLoad 16/600, GE Healthcare – 28-9893-35) in CSF-XB buffer and 250 mM sucrose. The purity of the proteins was confirmed by Coomassie-stained SDS-PAGE gels. Protein concentration was determined using an A280 Nanodrop (Thermo Scientific Nanodrop lite) with the corresponding extinction coefficient (calculated based on protein sequence and using ProtParam calculator https://web.expasy.org/protparam/). 10–20 μL protein aliquots were flash-frozen using liquid N₂ and stored at −80 °C.

In the importin testing system, both crude interphase extract and clarified extract were used. Clarified extracts were prepared as described[82]. Briefly, crude mitotic extracts were spun for a second time at 1,090,050 × g in a Beckman TLS-100A rotor for 2 h at 4 °C. The clear middle layer was extracted using a 22-gauge needle. Fresh interphase egg extract was made as described earlier.

In experiments using GST-importin-β, Pierce Glutathione Magnetic Agarose beads (Thermo Scientific 78602) were washed twice in XB buffer and then incubated in 1 h with purified GST-importin-β at 2 μg/μL proteins per μL beads. Flow-through importin β was removed post incubation to avoid free importin β competing with the immobilized β in extract. GST-importin-α was added at the ratio 1:1 importinα : importinβ molar concentration and incubated for 30 min. RanQ69L was titrated into the solution at estimated importin β :RanQ69L ratios of 1:0 to 1:100. Finally, extract was added to the mixture so that the final importin β concentration reached 10 μM and incubated for 1 h at 16 °C. After the incubation, beads were washed two times with XB buffer and eluted with sample buffer (Invitrogen NP0007) for a quick check with Coomassie gels (Invitrogen NW00100BOX) and eventually MS analysis.

In experiments using ZZ-tag-importin-β construct (expressed as described earlier), IgG Sepharose 6 Fast Flow (Cytiva 17-0969-01) were washed twice in XB buffer and then incubated with the cell lysate of the overexpressed construct for 1 h. The supernatant was then removed, and beads were washed once with XB buffer. RanQ69L was titrated into the solution at an estimated RanQ69L ratio of 1:0 to 1:100 and incubated for 30 min before removing the flow-through. GST-importin-β was added at the ratio 1:1 molar concentration. Finally, the fresh extract was added, and the pull-down collection was performed as described above.

The relative protein signals from the MS analysis were normalized by the added importin signal and IgG signals (in the case of the ZZ-tag-importin-β experiment). The protein fraction was defined as signal of importin-bound protein in a condition with RanQ69L divided by the sum signal of protein in the conditions with and without RanQ69L, reflecting the change in the RanQ69L amount in each condition. The protein fractions were then normalized using the values of known background proteins (such as glycolytic enzymes and mitochondrial

proteins) that do not interact with importin $\beta$. For each protein, the fractions were fitted through a linear function of the normalized added RanQ69L amount detected in the pull-downs. The y-intercept was fixed at 0.5. The extracted slope was used as a proxy for the protein's affinity to importin $\beta$. The experiments were repeated three times, and the measurements were projected onto a single dimension that maximized the agreement in variation between the $T_{embryo1/2}$ and the importin affinity proxy using canonical correlation analysis[49]. The projection is cross validated as following: The dataset was split into 10 consecutive folds with shuffling. At each fold, 90% of the data was used for training the canonical correlation axis while 10% was kept behind for validation. The validated values of the 10% portion from each round were collected from each round and made up the final vector of cross-validated projected values on the canonical axis. The projected values defined the final proxy for importin affinities and were used for downstream analysis.

### Ran-dependent importin affinity assay for a set of previously measured $K_D$ NLS sequences

We performed affinity assay for NLS peptide (Supplementary Fig. 8) sequences of previously measured $K_D$s by[50] spanning a wide range of affinities. With those, we performed the equivalent importin assay used for the proteome-wide affinity measurement (Fig. 4). To 1.5 μM of ZZ-tagged-importin-$\beta$, we added 150 nM of the tested NLS peptides. We then varied the RanQ69L molar ratio to importin $\beta$ from 1:0 to 1:100. After 1-hour incubation, we removed importin-$\beta$-covered beads together with bound peptides. We collected the supernatant, TCA precipitated to remove any remaining proteins, C18 purified peptides via staged tips and analyzed via label-free MS quantification. We repeated this experiment four times and required peptides to be measured at least in two replicates.

The MS files are processed on GFY (licensed from Harvard University) to identify the peptides and quantify total ion counts. The peptide signals were normalized the median signal across all peptides and conditions being quantified.

### DNA affinity assay

A pQE-80L empty plasmid (from SnapGene) was cut using EcoRI and BamHI (New England Biolabs) and purified using QIAquick Gel Extraction Kit (Qiagen). The collected DNA fragments were end-filled using Klenow (New England Biolabs) and biotin-dATP (Invitrogen). DNA was coupled to streptavidin Dyna beads (65305; Invitrogen) following the protocol outlined previously[83]. DNA bound to beads was collected at the estimated 1 μg/μL per beads volume and saved for future experiments.

Fresh interphase egg extract was made as described earlier. DNA beads were incubated in fresh extract for 2 h at concentrations ranging from 60 ng/μL to 160 ng/μL. Cycloheximide was added to arrest extract in the interphase. Post incubation, pull-downs were collected and eluted with 6 M Guanidine chloride pH 7.2 and subjected to MS analysis.

### Absolute protein concentration estimates

For absolute protein abundance estimates (Supplementary Data 4), we reanalyzed previously collected mass spectrometry data with a protein reference database based on the 9.2 version of the *Xenopus laevis* genome downloaded from Xenbase (http://ftp.xenbase.org/pub/Genomics/JGI/Xenla9.2/sequences/XENLA_9.2_Xenbase.pep.fa)[57,84–86]. The previously published estimates were analyzed with an mRNA reference database. To deduce the power-law relationship between MS-signal and protein concentration, we generated a regression linking the average log ion signal per peptide to pre-existing estimates of the protein concentration[57,84]. We measured the ion flux integrated over time by peptide, determined the total log ion signal for the pro-

tein, and divided by the total number of peptides in the protein to calculate the average log ion signal per peptide. We related the normalized log ion signal to the pre-existing protein concentration estimates using a robust regression since such data is often error-prone and skewed by outliers. We binned normalized log ion signals into bins of size 1/3 (in log ion signal space), calculated the median protein concentration across all proteins with normalized log ion signals in the range of the bin, and then fit a robust regression with a trimmed mean M-estimator with Ramsay's Ea of 1.65. We used that fit to estimate the protein concentration for all proteins detected in the *Xenopus* egg through mass spectrometry.

### Modelling of nuclear import in embryos and encapsulated droplets

We develop a model for nuclear import in early embryos based on the differential affinities of proteins to importin and the experimentally observed increase in total nuclear volume.

In this simple description, an embryo with a constant total volume ($V_{embryo}$) contains protein $i$, where $i = 1, \ldots, n$ for $n$ total proteins, with $P_i$ is protein abundance of protein $i$ with an embryonic concentration $[P_i] = \frac{P_i}{V_{embryo}}$. Protein $i$ is imported from the cytoplasm to the nucleus, where $[P_{cyto,i}] = \frac{P_{cyto,i}}{V_{embryo}}$ and $[P_{nuc,i}] = \frac{P_{nuc,i}}{V_{embryo}}$ represent embryonic concentration of protein $i$ located in the cytoplasm and nucleus respectively. Here, $P_{cyto,i}$ is the abundance of $i$ in cytoplasm and $P_{nuc,i}$ is in the nucleus. Protein $i$ undergoes the following transformation:

$$P_{cyto,i} \rightarrow P_{nuc,i} \qquad (1)$$

Note that $[P_{cyto,i}] + [P_{nuc,i}] = [P_i]$ to mass balance Eq. 1. If we denote the net flux of proteins from the cytoplasm into the embryonic nuclei as $F_{total}$, for a particular protein $i$ we can write the change in its abundance of those resided in the nucleus of the embryo as:

$$\frac{dP_{nuc,i}}{dt} = \theta_i F_{total} \qquad (2)$$

Where $\theta_i$ is the fraction of the total nuclear import flux contributed by protein $i$. The total nuclear import flux $F_{total}$ is the rate at which the total embryonic nuclear volume increases throughout development multiplied by the total protein concentration in the embryos.

$$F_{total} = \frac{dV_{nuc}}{dt} \sum_{i=1}^{n} [P_i] \qquad (3)$$

Based on experimental results, the total protein concentration $\sum_{i=1}^{n} [P_i]$ stays roughly constant throughout early development and is approximately ~2 mM (Supplementary Data 1)[21]. The rate of nuclear volume expansion, $\frac{dV_{nuc}}{dt}$ is written as the time derivative of the total nuclear volume as derived from our immunofluorescence data (Supplementary Fig. 9a). We assume that 70% of the cytoplasmic volume is excluded by yolk and lipids[29,87].

To derive $\theta_i$, the fraction of nuclear influx contributed by protein $i$, we assume that substrate binding to importin is in equilibrium, as in the Langmuir adsorption isotherm model[88].

$$P_{cyto,i} + I \underset{}{\overset{K_{Di}}{\rightleftarrows}} IP_{cyto,i} \qquad (4)$$

Where $[I]$ is the importin concentration in the embryo, which has a fixed total concentration $[I_0]$ throughout the considered developmental period[57], and $K_{Di}$ is the equilibrium dissociation constant for protein $i$ in this reaction. At equilibrium, $K_{Di}$ is:

$$K_{Di} = \frac{[P_{cyto,i}][I]}{[IP_{cyto,i}]} \qquad (5)$$

Using the mass balance equation

$$[I] = [I_0] - \sum_{i=1}^{n} [IP_{\text{cyto},i}] \qquad (6)$$

we can combine Eqs. 5 and 6 to find the fraction of total importin that is bound by protein $i$:

$$\theta_i = \frac{[IP_{\text{cyto},i}]}{[I_o]} = \frac{\frac{[P_{\text{cyto},i}]}{K_{Di}}}{1 + \sum_{j=1}^{n} \frac{[P_{\text{cyto},j}]}{K_{Dj}}} \qquad (7)$$

Combining Eqs. 2, 3, and 7, we arrive at a system of $n$ differential equations, where $n$ is the total number of proteins:

$$\frac{dP_{\text{nuc},i}}{dt} = \left( \frac{\frac{[P_{\text{cyto},i}]}{K_{Di}}}{1 + \sum_{j=1}^{n} \frac{[P_{\text{cyto},j}]}{K_{Dj}}} \right) \frac{dV_{\text{nuc}}}{dt} \sum_{k=1}^{n} [P_k] \qquad (8)$$

This system of $n$ differential equations and mass balance equations are solved using the ode45 function in MATLAB.

We created a synthetic system of nuclear proteins over a distribution of importin affinities in a background of importin-inert cytoplasmic proteins to mimic an actual embryo. We estimate that nuclear proteins constitute ~10% of the embryonic proteome mass based on the quantified NCV-ratio in the oocyte (Fig. 1b). Nuclear proteins' affinities to importin are sampled from a log-normal distribution, i.e. the natural logarithm of $K_{Di}$ are sampled from $\ln(K_{Di}) N(\text{mean} = -18, \text{standard deviation} = 2)$ which corresponding to a median affinity ($K_D$) of ~30 nM -a benchmark binding affinity measured for protein-protein interactions[89,90]. Cytoplasmic proteins account for 90% of the proteome and have no affinity to importin ($K_{Di} = \infty$). We estimate the protein concentration for each species from the total protein abundance estimation in the egg[57]. A median protein concentration is ~44 nM and the total protein concentration is ~2 mM in the egg (Supplementary Data 4)[57].

We simulated embryonic nuclear import with 4,600 nuclear proteins, each at a concentration of 44 nM representing 10% of the proteome, and cytoplasmic proteins at 1.8 mM concentration, and 1.5 µM of importin (constant over time). For an illustration of differential nuclear entry due to importin affinities, two representative proteins (one with high importin affinity (10 nM) and one with low importin affinity (1 µM)) show the expected differential nuclear entry times into the increasing embryo's nuclear volume (Fig. 3e). For an illustration of sequential titration of nuclear concentration due to the continuous nuclear import over development, we simulate three representative proteins: a high-affinity protein (1 nM), an intermediate affinity (30 nM), and a low affinity (1 µM).

Similarly, we test the model in a single cell cycle after the nuclear envelope reformation. Compared to the embryo simulations, we adapted a modification on nuclear flux: The nuclear flux, $\frac{dV_{\text{nuc}}}{dt}$ is derived from the change of the nuclear volume in cell-free droplets (Supplementary Fig. 9c). We simulated a similar system in early development with the adjusted nuclear flux and illustrated that the nuclear concentration of titrating proteins from 1 nM, 30 nM, to 1 µM sequentially reaches the maximum as the nucleus grows in cell droplets (Fig. 4e).

### Assaying nuclear import in oil encapsulated artificial cells

The Gateway entry plasmids of desired proteins were retrieved from *Xenopus laevis* ORFeome[91]. The destination vector carrying an EGFP sequence−TEV site−S-tag (pCSF107mT-GATEWAY-3'-LAP tag) was chosen and bought from Addgene. For the Gateway LR cloning reaction, the entry plasmid, the destination plasmid, and the Gateway LR clonase II enzyme mix (Invitrogen 11791) were combined at the ratios recommended in the manufacture protocol. After the reaction, the expression cloned vector was purified, then linearized using restriction enzymes, which were chosen so that the region of protein of interest was protected. The linearized plasmids were in-vitro transcribed using the mMESSAGE mMACHINE SP6 kit (Invitrogen AM1340) supplemented with a 7-methyl guanosine cap protected on the 5' end terminal, and a poly(A) tail (NEB M0276). Finally, RNA products were purified using Trizol LS reagent (Invitrogen 10296010), then resuspended in nuclease-free water at ~1 µg/µL in the final RNA concentration.

We microinjected the RNA products into the oocytes using a PM2000B 4-channel Pressure Injector (MicroData Instrument). ~100 oocytes per protein construct were injected twice to be equally distributed around the animal cap at the total volume of ~50 nL. Injected embryos were allowed to recover in 2.5% Ficoll OCM and visually inspected before use in all experiments. After 1-hour resting in Ficoll, oocytes were transferred to OCM for overnight expression. Only healthy oocytes were used the next day to make the extract. Oocyte extract was collected as previously described. Finally, the undiluted extract was supplemented with 50 mM sucrose, 20 µg/mL LPC protease inhibitors, and 20 µg/mL cytochalasin D. The presence of the desired protein was validated via epifluorescence imaging of anti-S-tag beads pulled down from the extract (SinoBiological MB101290-T38). After confirming the expression, the extract was flash-frozen and stored in ~1 µL aliquots at −80 C for further use.

CSF-arrested *Xenopus laevis* egg extract was driven into interphase by supplementing $Ca^{2+}$ [4 mM] and demembranated sperm nuclei[92], which served as chromatin sources for nuclear assembly, to a final concentration of 1E6 nuclei/µL. To facilitate imaging of growing nuclei, mCherry GST-NLS 11.5 mg/mL was added to the master mix at a 1:100 dilution. Candidate GFP-protein conjugates were also added to the extract mix at 1:25 dilution prior to nuclei encapsulation in droplets, with concentrations ranging from 0.5 mg/mL to 3.0 mg/mL depending on the protein being used.

Polydimethylsiloxane (PDMS) T-junction microfluidic devices affixed to #1.5 coverslips were used to generate monodisperse emulsions of ~50 µm diameter extract droplets in a continuous oil phase as previously described[93]. To facilitate the use of small extract volumes, 2 mL Eppendorf tubes containing 25 µL of extract were placed on ice and pressurized using microfluidic pressure pumps (Flow EZ 1000, Fluigent). Flowrates of both the extract and oil phases were controlled by modulating the applied pressures. PDMS devices were kept on ice during filling. Once a device was filled, its inlet and outlet channels were sealed with silicon tubing plugs and the device was then placed on the microscope stage for imaging.

All imaging was performed in a temperature-controlled room at 18 °C. Image acquisition was conducted using an IX-81 confocal microscope (Olympus USA) equipped with a 40×0.6NA objective and a CSU-W1 confocal scanning unit. Images were captured with an ORCA-Flash4.0 sCMOS camera (Hamamatsu Photonics) and system automation was controlled via cellSens software (Olympus USA). Time-lapse image series of protein import into encapsulated nuclei were generated by acquiring z-stacks (step-size = 2 µm, 12 slices per time point) in both the red and green channels at 2 min intervals. Acquisition began 5–6 min after filled devices were removed from ice (we called this $t = 0$) and typically lasted for 1–2 h.

The collected image series were analyzed using Fiji software (NIH). Once collected, hyperstacks were parsed by channel and a modified version of the Autofocus hyperstack macro (Richard Mort) was used to extract a single, best-focused image of the nucleus for each time point and generate a time-lapse series of nuclear import. Before proceeding with analysis, automated best-focused selections were then confirmed manually. Images were typically segmented using the mCherry signal as a reference to generate ROIs outlining the nucleus. In the rare case in which the GFP-protein entered the nucleus first, the GFP channel was used as a reference to generate ROIs. These

ROIs were then copied to corresponding images in the opposing channel and the mean and integrated fluorescence intensities of the nuclear ROIs were measured for each channel, red and green, using the Analyze Particles function in Fiji. To measure cytoplasmic fluorescence intensities, nuclear ROIs were dilated, and the same measurements repeated. Cytoplasmic intensity was then calculated by subtracting the integrated intensity of the original nuclear ROI from the dilated ROI integrated intensity. The relative nuclear concentration was calculated by dividing the mean nuclear intensity by the sum of the mean nuclear and cytoplasmic intensities. Since the extract was well-mixed and uniform before the formation of the nucleus, the RNC at the initial time point was 0.5. The RNC data were fitted with a sigmoid function to extract the time $T_{\text{droplet 1/2}}$, at which the relative intensity reaches half of its max value for each protein and the corresponding mCherry-NLS. To overcome extract variability, the import time difference ($\Delta T_{\text{droplet 1/2}}$) between mCherry-NLS and the GFP-conjugated protein of interest was used to compare the nuclear import rates between proteins of interest.

### Assaying nuclear import of various NLS-GFP fusions

We fused the bioinformatically predicted NLS sequence from proteins assayed in Fig. 5, (Yy1, Gtf3a, Gtf2h1) next to GFP in a plasmid for bacterial expression[51]. We expressed and purified NLS$_{\text{Yy1}}$-GFP, NLS$_{\text{Gtf3a}}$-GFP, and NLS$_{\text{Gtf2h1}}$-GFP at ~1.5 μM concentration. We added the expressed proteins and 1.5 μM NLS$_{\text{SV40}}$-mCherry to *Xenopus* egg extracts doped with sperm DNA. We collected samples every five minutes fixed with Hoechst from 15-minute post sperm addition to 120-minute post sperm addition. We imaged nuclear entry of mCherry and GFP with confocal microscopy.

We processed and quantified the images using ImageJ. Images were typically segmented using the mCherry signal and verified by the presence of nuclear signal via either Hoechst (for early time points) or GFP (for late time points). The segments are references to generate ROIs outlining the nucleus. At least three nuclei are quantified for each collection time point. Once nuclear ROIs are defined, we measure the relative nuclear and cytoplasmic intensity and fit a sigmoid function to extract the $T_{1/2}$, at which the relative intensity reaches half its max value for each protein in each experiment.

### Reporting summary

Further information on research design is available in the Nature Research Reporting Summary linked to this article.

## Data availability

The data that support this study are available from the corresponding author upon reasonable request. The mass spectrometry proteomics data have been deposited to the ProteomeXchange Consortium via the PRIDE partner repository[94] with the dataset identifiers PXD028069 and PXD036403. Proteomics data analysis was performed using GFY Core version 3.8, Python version 3.7.9, and MATLAB version 2018b. Analyzed proteomics data are provided in the Supplementary Data files 1–4 and the Source Data file. Immunofluorescence imaging data and fluorescence cell droplet data were acquired on Zeiss 880 confocal microscope and IX-81 Olympus confocal microscope respectively. Imaging data was analyzed using Fiji software. (NIH) and Python version 3.7.9. Analyzed imaging data are available via the Source Data file. Source data are provided with this paper.

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

## Acknowledgements

We thank Matt Sonnett, Eyan Yeung, Lillia Ryazanov, Nick Treen for help and training and Eric Wieschaus, Mike Levine, Elizabeth Van Itallie, and members of the Wühr Laboratory for their comments and edits on the manuscript. We thank David Hill for access to the *Xenopus* ORFeome and Dirk Görlich, Thomas Güttler, and Sabine Petry for the gifts of RanQ69L and importin plasmids. We thank James Pelletier for his help with designing the filter holders. Lastly, we thank John Oakey and Cassidy Enloe for their help with PDMS device design and manufacture. This work was supported by NIH grant R35GM128813 (M.W.), P20-GM113132 (A.A.), R01GM135568 (J.G.), American Heart Association predoctoral fellowship 20PRE35220061 (T.N.), National Science Foundation Graduate Research Fellowship (E.R.C.), EMBO ALTF 601-2018 (F.C.K.), Princeton Catalysis Initiative (M.W.), Eric and Wendy Schmidt Transformative Technology Fund (M.W.), Harold W. Dodds Fellowship (M.G.), Princeton University's Summer Undergraduate Research Program (E.C., J.R., C.K.), NSF MODULUS award 2052640 (J.G.).

## Author contributions

M.W. and T.N. conceptualized the study. T.N. and E.C. performed importin affinity experiments. T.N. and J.R. performed DNA affinity experiments. T.D., M.T., and T.N. performed in vitro nuclear entry experiments with JG's supervision. T.N. and F.C.K. analyzed imaging data. T.N., M.S., and E.R.C. generated and analyzed mass spectrometry data. T.N. and M.G. developed the statistical framework. C.K.K. analyzed absolute abundance data. M.W., A.A., and J.G. provided funding. M.W., J.G., and A.A. supervised the study. T.N., M.W., and E.C. wrote the manuscript, and all authors helped edit the manuscript.

## Competing interests

The authors declare no competing interests.
