## [Peer Review File · Nature Communications]

REVIEWER COMMENTS

Reviewer #1 (Remarks to the Author):

The manuscript by Nguyen et al. describes the kinetics of nuclear protein import during early embryogenesis in *Xenopus*. They provide an explanation for the ordered activation of transcriptional processes during the first cell divisions until a stage of several thousand cells. They postulate that differential affinity of nuclear proteins to the classical nuclear import machinery consisting of importin alpha and beta decides about the speed of their import. To this purpose they developed an elegant proteomics based assay to study the kinetics of nuclear import and clearly show that some proteins enter the nucleus earlier than others and that this correlates well with their known sequential transcriptional activities. These results are convincing but somehow expectable. However, the postulated mechanism based on the affinity to importins, which are present in limiting amounts, needs more evidence to be accepted, since it only relies on correlations:

- 1) *Xenopus*, like mammals, has at least 6 alpha-importins and at least 3 of them have been detected in oocytes (Levy and Heald Cell 2010 PMID: 20946986). The authors only use one, KPNA2, for the affinity assays, which is too simplistic. Moreover, KPNA2 is dynamically phosphorylated during mitosis, needs phosphorylation to support nuclear import, and its concentration declines during embryogenesis in *Xenopus* (Levy & Heald, Cell 2010 PMID: 20946986), which probably is not taken into account in the modelling of the authors. In general, the authors do not comment on the appearance of different importins in the nucleus at different time points; these are important data to provide.
- 2) Moreover, the proteins shown in Fig. 3 employ very different import machineries, SMAD3 uses only a beta-importin, SMAD4 an alpha/beta complex (Hill, Cell Res 2009, PMID: 19114992), SMARCA4 uses a KPNA6/beta complex (Hügel, Mol Cell Proteomics 2014, PMID: 24623588) and CTNNB1 enters the nucleus without import carrier (Fagotto, Curr Biol 1998, PMID: 9501980). YY1 tested in the droplet assay (Fig. 5) uses KPNA4/beta (Neira, BBA 2021, PMID: 33945888). This again shows that the importin binding assay using only KPNA2 is not reflecting the true affinity of the proteins to the import machinery.
- 3) Phosphorylation of proteins is a very important regulator of their nuclear import efficiency (Jans & Hübner, Physiol Rev 1996, PMID: 8757785), also in *Xenopus* embryos (Smillie J Cell Sci 2004, PMID: 15075245). Did the authors differentially analyze the nuclear and cytoplasmic phosphoproteomes?
- 4) The authors also ignore nuclear export kinetics as being relevant for the nuclear appearance of proteins (e.g., Kirli eLife 2015 PMID: 26673895).
- 5) The authors exclude DNA affinity to be an important parameter for nuclear import efficiency, but use artificial vector DNA for the binding assay. Most transcriptionally active proteins bind to specific sequences in the DNA or to other proteins which have such a specific DNA landing site. Thus, the importance of these specific interactions and the true affinity of each protein to the genomic DNA are not being assessed in the paper.

6) mCherry-NLS is very quickly imported. This NLS seems to be a high-affinity binder to importins. By varying it with other known NLS of proteins with slower import kinetics, the authors should test their hypothesis in a very convenient and convincing way.

7) Another essential test would be to increase the concentration of importins in the cytoplasm. This may be technically ambitious in the intact oocytes, but for sure doable in the droplet assay. If importins are limiting, as postulated by the authors, there should be a level of importins at which the speed of import of different cargoes gets equalized.

8) Fig. 5: Which of the nine proteins used in the droplet assay is shown in Fig. 5B? Fig. 5C shows only the order of appearance in the nucleus in both assays, not T_{1/2}. Please blot the measured numbers to get a reliable correlation.

Reviewer #2 (Remarks to the Author):

As early embryonic divisions create more and more nuclei, an increasing fraction of maternally deposited protein is imported into the nucleus. In this study, the authors use proteomic mass spec to monitor the nuclear proteins over time during early embryo development to determine whether some proteins are imported into the nucleus before others. After establishing that the import of nuclear proteins is ordered, the authors highlight how their measurements are consistent with what is already known. Finally, they pursue a mechanistic explanation for their measurements by assessing the importin binding affinities.

Overall, this is a very elegant use of proteomics. The data in this manuscript is presented clearly, and the results of the study are interesting and sensible. I recommend publication if the following points are addressed.

Major point:

From what I can tell, the authors only performed the primary experiment of this manuscript 1 time (though there are some columns in the supplement with the number "2" from non-overlapping timepoints). The authors must perform at least one independent biological replicate and then correlate the T_{1/2}s calculated from each experiment to demonstrate the degree of reproducibility. It would also be important to visualize where the individual proteins highlighted in the manuscript lie within the correlation plot. This replicate analysis may also help improve the correlation between the import affinity and T_{1/2}.

Minor points:

- 1) Are there predictions for the strength of a nuclear import sequence? Do proteins containing clear consensus NLS sequences among the first imported into the nucleus?
- 2) Do the binding affinities of the individual proteins highlighted in Figures 2 and 3 agree with the overall model. It would be nice to revisit those proteins in Figure 4.
- 3) Since it is such a striking observation, I encourage some speculation about the potential implications for the ordered import of the DNA repair proteins. Is it known whether certain DNA repair mechanisms are preferred over others during embryogenesis?
- 4) The graphical abstract is confusing. It is unclear what the check marks are supposed to denote.

Reviewer #3 (Remarks to the Author):

The manuscript under review provides evidence for the attractive proposal that the relative affinities of proteins for their cognate importins is a major determinant in the timing of protein entry into embryonic nuclei. If correct, this is an important finding with major implications for our understanding of nucleocytoplasmic transport. Although the experimental evidence, with its emphasis on quantitative proteomics, is of high quality, there is a control that I believe is needed to test the validity of the lysate MS affinity assay - i.e., testing the assay with a set of NLS reporters of known affinity. I emphasize that this test is needed because the hypothesis rides on the integrity of the affinity assay.

Reviewer #4 (Remarks to the Author):

The work by Nguyen et al sets up a methodology for systematically measuring time resolved nuclear import during embryonic development. They obtain quantitative temporal import measurements for close to 2000 nuclear proteins and convincingly illustrate that the import dynamics can to a significant extent explain the timing of transcriptional events during embryo development. Further they show that this sequential import can be rationalized in part by affinities of different nuclear proteins to importin. Finally, they set up an invitro assay to validate these findings.

This is beautiful work on many levels. The development of the improved nuclear purification protocol is impressive in terms of its performance and (Extended Data Figure 2). The temporal data is a very important resource that will be used by others in the field. Also, the biological findings and interpretation of the data are novel and exciting. I strongly recommend publication following minor revisions.

Minor comments:

For the NPC data: currently there is an extended figure comparing the basket proteins to other NPCs. As NPC is composed of several subcomplexes it would be interesting to illustrate and compare how different NPCs subcomplexes align or differ in their import dynamics.

The manuscript mentions that often proteins part of the same complex are co-imported. This would be nice to represent more systematically eg by using the Corum annotation and checking the significance of co-import of complexes vs random assignment (for similar type of analysis for protein turnover you could look up this paper Mathieson et al Nat Comm 2018).

I would also appreciate some discussion on proteins that are getting imported but do not have nuclear annotation.

Kind regards,

Mikhail Savitski

Reviewer #1 (Remarks to the Author):

The manuscript by Nguyen et al. describes the kinetics of nuclear protein import during early embryogenesis in *Xenopus*. They provide an explanation for the ordered activation of transcriptional processes during the first cell divisions until a stage of several thousand cells. They postulate that differential affinity of nuclear proteins to the classical nuclear import machinery consisting of importin alpha and beta decides about the speed of their import. To this purpose they developed an elegant proteomics based assay to study the kinetics of nuclear import and clearly show that some proteins enter the nucleus earlier than others and that this correlates well with their known sequential transcriptional activities. These results are convincing but somehow expectable.

We appreciate that Reviewer #1 is convinced of the key findings of our study. We would like to point out that the results obtained were not apparent to us before this study and were not evident to other experts in the field, as exemplified in a very recent review by (Blitz and Cho, 2021):

"In summary, while clearly many TFs do access nuclear chromatin before ZGA, the timing of translocation of some may be a regulated step. For most, the role of such regulation of nuclear translocation in controlling the timing of transcriptional activation remains to be established. A time-course using quantitative proteomics would be valuable to explore the temporal dynamics of TF protein expression levels, and proteomics on nuclei (Amin et al., 2014; Peshkin et al., 2015, Peshkin et al., 2019; Wühr et al., 2014, Wühr et al., 2015) would be critical to providing a global view of nuclear localization of the entire TF repertoire in the buildup to ZGA."

However, the postulated mechanism based on the affinity to importins, which are present in limiting amounts, needs more evidence to be accepted, since it only relies on correlations:

- 1) *Xenopus*, like mammals, has at least 6 alpha-importins and at least 3 of them have been detected in oocytes (Levy and Heald Cell 2010 PMID: 20946986). The authors only use one, KPNA2, for the affinity assays, which is too simplistic. Moreover, KPNA2 is dynamically phosphorylated during mitosis, needs phosphorylation to support nuclear import, and its concentration declines during embryogenesis in *Xenopus* (Levy & Heald, Cell 2010 PMID: 20946986), which probably is not taken into account in the modeling of the authors.

We fully agree with Reviewer #1 that our proposed model considering one importin 13/ α pair must be simplistic compared to what is truly happening in the developing embryo. The simplification of our model is an overarching criticism of Reviewer #1. However, like many reductive approaches, this simplification is a starting point that allows us to progress toward unraveling the complexity of embryonic development. We will attempt to address this general criticism of simplicity here and follow up with specific details in the following responses. It is arguably impossible, at least for the foreseeable future, to fully address the complexity mentioned by Reviewer #1, including substrate importin/exportin interactions, which include appreciable redundancy (Kimura et al., 2017; Wing et al., 2022), the modulation of importin affinities for multiple adaptors (importin α s), changes of expression levels of all proteins involved, and/or their post-translational modifications. To simplify this complex system, we chose to investigate importin 13 (Kpnb1) because it is the most abundant importin (Extended Data Fig. 6) and Kpna2 because it is importin 13's canonical adaptor (Lott and Cingolani, 2011). Nevertheless, even with these simplifications, our data and model explain >40% of the variance in the timing of nuclear import in the early frog embryo (Fig. 4), though we acknowledge that other import mechanisms most likely provide additional regulation layers and are likely essential for proper embryonic development. However, the characterization of these additional layers of regulation is beyond the scope of this paper and will likely require many future studies and decades of additional research to resolve. Despite the reductive nature of our study, we believe it represents remarkable progress: To our knowledge, the timing of nuclear entry into embryonic nuclei has not been considered to be a relevant timing mechanism in early development (e.g., see review by Blitz & Cho Curr Top Dev Biol 2021). We have adapted the text in the manuscript to further stress (i) the need for simplification in our model and (ii) that the proposed model does not fully represent the complexity that must occur in the embryo:

"By necessity, our model ignores various layers of nuclear import regulation, which are most likely crucial for embryonic development, including substrate interactions with the

other importins, exportins, and importin α s (Kimura et al., 2017; Wing et al., 2022), changes of expression levels of all proteins involved, and post-translational modifications of substrates or importins and exportins. Nevertheless, our ability to explain at least 48% of the observed variance for the timing of nuclear entry from these two simple assays is remarkable, especially considering that nuclear import in early embryos is undoubtedly more complicated than implied with these assays."

Additionally, we updated the text in the discussion to stress further that our model is simplified compared to the complexity of nuclear import in the early embryo:

"In addition, over 40% of the observed time-variance in nuclear import across all nuclear proteins was explained via differential affinities of the proteins for importin, despite quantifying affinities to only importin $\alpha/13$ using a crude and noisy biochemical assay. Obviously, regulation of nuclear transport in the embryo must be much more complicated than implied by our simplistic assay and model. Nevertheless, the observed predictive power suggests that this fundamental biochemical mechanism plays a crucial role in the temporal organization of developing early embryos that could set the stage before gene regulatory networks orchestrate cellular differentiation."

To clarify our reasoning for choosing the specific importin $\alpha/13$ in our model and provide background information on the changing levels of importins/exportins throughout early development, we added Extended Data Fig. 6 to the manuscript.

Extended Data Figure 6. The expression levels and subcellular localization of nuclear transport receptors observed in the frog embryos and their dynamic changes in early development.

a, The absolute abundance of transport receptors in the frog egg (Supplementary Table S4 and (Wuhr et al., 2015)). Among these, importin 13 (Kpnb1) is the most abundant protein.

b, Relative protein abundance (left) and the subcellular localization (right) of the importin-13 family importins as a function of developmental progression. We observed that the expression levels of importins stay constant throughout development and that the proteins locate preferentially in the cytoplasm throughout early development. The preferential cytoplasmic partition of importins was similarly observed in the frog oocyte (Wuhr *et al.*, 2015).

c, Relative protein abundances of importin α s in early frog embryos show that their levels remain approximately constant.

d, Relative protein abundance changes of the exportins and biportins as a function of developmental progression. Most exportins and biportins remain approximately constant. The exception is exportin 6 (Xpo6). Xpo6 primary substrate is actin. Xpo6 is absent in *Xenopus* oocytes, which results in nuclear actin localization that supports the physical integrity of the large oocyte nucleus (Bohnsack et al., 2006; Stuken et al., 2003). Upon fertilization, Xpo6 expression level increases, and actin is excluded from embryonic nuclei.

Our model is agnostic to the underlying changes of the molecular mechanism that affect the total import rate (e.g., protein phosphorylation or changes of importin abundance), as we derive this rate empirically from the changes in nuclear volume via immunofluorescence imaging (Extended Data Fig. 9a, Supplementary Material, and Methods). Additionally, we would like to clarify that Wilbur and Heald 2013 indicated that the total importin α level does not change throughout development. Instead, a fraction of α , which is cytoplasmic at the early stage, is incorporated into the cell membrane at later stages (Brownlee and Heald, 2019; Levy and Heald, 2010; Wilbur and Heald, 2013). "Although previous data showed that cytoplasmic importin α levels decreased by stage 8 of *Xenopus* development (Levy and Heald, 2010), immunoblotting of importin α in whole-embryo lysates, rather than cytoplasmic extracts, revealed that importin α partitions into multiple populations with no apparent decrease in the total amount." (Wilbur and Heald, 2013). Consistently, this study and previous publications from our laboratory and others observe a constant total expression of importin α during early

embryogenesis (Extended Data Fig. 6c). (Peshkin et al., 2019; Peshkin et al., 2015; Van Itallie et al., 2021).

Based on the reviewers' comments we updated the text to incorporate the discussion of membrane segregation of importin α levels and added the speculation that this could explain the observed decrease of total nuclear import rate after the ZGA (Extended Data Fig. 9a). We added the following text to the figure caption:

"The maximum volume during this time normalizes the nuclear volume (y-axis left), and the nuclear flux is normalized by the maximum flux (y-axis right). The decline in the total nuclear import rate might be due to the previously reported sequestration of importin α to the cellular membrane (Brownlee and Heald, 2019)."

2) In general, the authors do not comment on the appearance of different importins in the nucleus at different time points; these are important data to provide.

A description of the change in concentration of all measured nuclear proteins as a function of the time is available in the original Supplementary Table 1,2. To highlight the particular data points Reviewer #1 requests, we added a new Extended Data Fig. 6b, c, d showing the change of protein abundances of importins and exportins as a function of developmental progression and the uptake of importins into the nuclei of developing embryos.

Extended Data Figure 6. The expression levels and subcellular localization of nuclear transport receptors observed in the frog embryos and their dynamic changes in early development.

a, The absolute abundance of transport receptors in the frog egg (Supplementary Table S4 and (Wuhr et al., 2015)). Among these, importin 13 (Kpnb1) is the most abundant protein.

b, Relative protein abundance (left) and the subcellular localization (right) of the importin-13 family importins as a function of developmental progression. We observed that the expression levels of importins stay constant throughout development and that the proteins locate preferentially in the cytoplasm throughout early development. The preferential cytoplasmic partition of importins was similarly observed in the frog oocyte (Wuhr *et al.*, 2015).

c, Relative protein abundances of importin α s in early frog embryos show that their levels remain approximately constant.

d, Relative protein abundance changes of the exportins and biportins as a function of developmental progression. Most exportins and biportins remain approximately constant. The exception is exportin 6 (Xpo6). Xpo6 primary substrate is actin. Xpo6 is absent in *Xenopus* oocytes, which results in nuclear actin localization that supports the physical integrity of the large oocyte nucleus (Bohnsack et al., 2006; Stuken et al., 2003). Upon fertilization, Xpo6 expression level increases, and actin is excluded from embryonic nuclei.

The modest enrichment of importins in the cytoplasm is consistent with our previous findings that importins are preferentially located in the cytoplasm while exportins are preferentially located in the nucleus, with biportins being close to equidistributed. Please see Supplementary Fig S3 from our previous paper (Wuhr *et al.*, 2015).

Supplementary Figure S3C from (Wuhr *et al.*, 2015)) Estimated concentrations and relative nuclear localization of nuclear transport receptors found in the frog oocyte.

- 3) Moreover, the proteins shown in Fig. 3 employ very different import machineries, SMAD3 uses only a 13-importin, SMAD4 an α /13 complex (Hill, Cell Res 2009, PMID: 19114992), SMARCA4 uses a KPNA6/13 complex (Hügel, Mol Cell Proteomics 2014, PMID: 24623588) and CTNNB1 enters the nucleus without import carrier (Fagotto, Curr Biol 1998, PMID: 9501980). YY1 tested in the droplet assay (Fig. 5) uses KPNA4/13 (Neira, BBA 2021, PMID: 33945888). This again shows that the importin binding assay using only KPNA2 is not reflecting the true affinity of the proteins to the import machinery.

As discussed above, we fully agree that our model provides an overly simplistic description of what is occurring in the embryo. We have discussed this in greater detail in response to Reviewer #1's first comment. As already mentioned above, we have updated the text in the manuscript and the discussion section to make the rationale for our necessary reductionist approach more explicit:

"By necessity, our model ignores various layers of nuclear import regulation, which are most likely crucial for embryonic development, including substrate interactions with the other importins, exportins, and importin α (Kimura *et al.*, 2017; Wing *et al.*, 2022), changes of expression levels of all proteins involved, and post-translational modifications of substrates or importins and exportins. Nevertheless, our ability to explain at least 48% of the observed variance for the timing of nuclear entry from these two simple assays is remarkable, especially considering that nuclear import in early embryos is undoubtedly more complicated than implied with these assays."

"In addition, over 40% of the observed time-variance in nuclear import across all nuclear proteins was explained via differential affinities of the proteins for importin, despite quantifying affinities to only importin α/β using a crude and noisy biochemical assay. Obviously, regulation of nuclear transport in the embryo must be much more complicated than implied by our simplistic assay and model. Nevertheless, the observed predictive power suggests that this fundamental biochemical mechanism plays a crucial role in the temporal organization of developing early embryos that could set the stage before gene regulatory networks orchestrate cellular differentiation."

- 4) Phosphorylation of proteins is a very important regulator of their nuclear import efficiency (Jans & Hübner, *Physiol Rev* 1996, PMID: 8757785), also in *Xenopus* embryos (Smillie *J Cell Sci* 2004, PMID: 15075245). Did the authors differentially analyze the nuclear and cytoplasmic phosphoproteomes?

We thank the Reviewer for raising this suggestion, but despite our expertise in phosphoproteomics (e.g., Presler et al. *PNAS*, 2017), assaying the change of phosphorylation sites as a function of nucleocytoplasmic partitioning in the early embryo is currently not feasible experimentally. Phosphoproteomics requires 10x more material, which is not achievable using our current approach for nuclear enrichment (Fig. 2a). It took us years to establish the assay for nucleocytoplasmic partitioning on the protein level. It would likely take several months of optimization before we could perform a phosphoproteomics experiment, much longer than the time we have to respond to these reviews. Even if we could reasonably acquire such data, it is unclear to us how we would incorporate them in our analysis and/or modeling, and we believe such data would not significantly improve the manuscript.

We have added a point in the discussion, as quoted previously in our response to comment 1, mentioning the role of phosphorylation in regulating nuclear import.

- 5) The authors also ignore nuclear export kinetics as being relevant for the nuclear appearance of proteins (e.g., Kirli *eLife* 2015 PMID: 26673895).

We agree that we are ignoring important additional layers of regulation. Unlike importins and importin α s, one exportin (Exportin 6) changes its expression levels significantly. We added this to the supplement (Extended Data Fig. 6).

Extended Data Figure 6. The expression levels and subcellular localization of nuclear transport receptors observed in the frog embryos and their dynamic changes in early development.

a, The absolute abundance of transport receptors in the frog egg (Supplementary Table S4 and (Wuhr *et al.*, 2015)). Among these, importin 13 (Kpnb1) is the most abundant protein.

b, Relative protein abundance (left) and the subcellular localization (right) of the importin-13 family importins as a function of developmental progression. We observed that the expression levels of importins stay constant throughout development and that the proteins locate preferentially in the cytoplasm throughout early development. The preferential cytoplasmic partition of importins was similarly observed in the frog oocyte (Wuhr *et al.*, 2015).

c, Relative protein abundance of importin α s in early frog embryos show that their levels remain approximately constant.

d, Relative protein abundance changes of the exportins and biportins as a function of developmental progression. Most exportins and biportins remain approximately constant. The exception is exportin 6 (Xpo6). Xpo6 main substrate is actin. Xpo6 is absent in *Xenopus* oocytes, which results in nuclear actin localization that supports the physical integrity of the large oocyte nucleus (Bohnsack *et al.*, 2006; Stuken *et al.*, 2003). Upon fertilization, Xpo6 expression level increases, and actin is excluded from embryonic nuclei.

The main substrate of exportin 6 (Xpo6) is believed to be actin. The change of the expression level is consistent with high actin levels inside the nucleus in the mature oocyte while actin gets excluded from nuclei in the early embryos (Bohnsack *et al.*, 2006; Clark and Merriam, 1977). We add a figure below showing our observation is consistent with the expectation of actins' subcellular localization in the oocyte and early embryos.

It is known that Xpo6 exports β -actins, α -actins, and gamma-actins (Bohnsack *et al.*, 2006). Here, we also observed that gamma actin (Actg1) and α -actin (Act3) are present in the oocyte nucleus while localizing completely cytoplasmic in the early embryos.

Additionally, we would like to point out that there is some evidence in the literature that inhibition of exportin functionality has a surprisingly small effect on early development. The highest expressed exportin with a broad spectrum of substrates is Xpo1 (The higher expressed Cse11 main functionality is to export importin α . (Guttler and Gorlich, 2011; Kutay *et al.*, 1997; Mackmull *et al.*, 2017)). Interestingly, Callanan *et al.* 2000 have shown that the embryo does not show an apparent phenotype to the inhibition of exportin 1 with the highly selective and potent inhibitor leptomycin B until reaching the initiation of the neurula stage (~8 hours after the ZGA at 16 degrees C) (Callanan *et al.*, 2000).

6) The authors exclude DNA affinity to be an important parameter for nuclear import efficiency, but use artificial vector DNA for the binding assay. Most transcriptionally active proteins bind to specific sequences in the DNA or to other proteins which have such a specific DNA landing site. Thus, the importance of these specific interactions and the true affinity of each protein to the genomic DNA are not being assessed in the paper.

We show that measured DNA affinity seems to contribute (albeit modestly) to predicting the timing of nuclear affinity (Extended Data Fig. 8). Although we agree with the

Reviewer that a better assay would have been to use frog DNA rather than plasmid DNA for the DNA affinity measurements, such an assay was infeasible. We initially attempted to set up this assay based on previously published frog-DNA covered magnetic beads (Guse et al., 2012). However, those assays that were developed for imaging purposes needed only minimal amounts of beads for microscopy purposes. Comparatively, our proteomics assays require orders of magnitude more material, and we were not able to generate sufficiently large amounts using frog sperm DNA. We were only able to obtain sufficient material for the proteomics assay when using plasmid DNA. We agree that the Reviewer raises an important point that some protein-DNA affinity might not be captured with the usage of generic DNA and have added the following text to the manuscript and extended figure legend:

"Differential protein affinities to either DNA or importins might result in ordered nuclear import. We found that plasmid DNA affinity was poorly predictive of nuclear entry time (Extended Data Fig. 8a, b)."

"The correlation suggests that plasmid DNA affinity explains 14% of the nuclear import variance in the embryos. For technical reasons, we could not perform these assays with frog DNA. Therefore, this assay does not capture protein affinity to frog-specific DNA sequences."

7) mCherry-NLS is very quickly imported. This NLS seems to be a high-affinity binder to importins. By varying it with other known NLS of proteins with slower import kinetics, the authors should test their hypothesis in a very convenient and convincing way.

Thanks for this suggestion. As the Reviewer suggested, we varied the NLS on the fluorescent protein with the NLSs of known proteins with slower import kinetics. To this end, we expressed GFP with the predicted nuclear localization signals from Yy1 (a fast-imported protein), Gtf2h1, and Gtf3a (slow-imported proteins) and performed nuclear import assays. As predicted, we observe that GFP enters the nuclei rapidly when we attach the Yy1 sequence; this import is even faster than with the standard SV40 NLS. In contrast, using either the predicted signal of Gtf3a or Gtf2h1 reduces the import rate of GFP, making it slower than the standard SV40 NLS, hence much slower than Yy1. We added the figure below, which summarizes the results of this assay.

a proteins: predicted NLS sequences —O

control: SV40 NLS — mCherry | Gtf3a: GGRMKKGGGGSGKSSKKS —

Yy1: KRKLKEKCPRPKR — GFP | Gtf2h1: LPKFKRKANKELEEKNR — GFP

Extended Data Figure 10: Import kinetics of NLS-GFP fusions correlate with import kinetics of the NLS-origin proteins.

a, We transferred the bioinformatically predicted NLS sequence from proteins assayed in Fig 5, Yy1 (a fast-imported protein $AT_{\text{droplet } 1/2} = -15.0$ min) and the slow-imported proteins Gtf2h1 ($AT_{\text{droplet } 1/2} = 41.1$ min), and Gtf3a ($AT_{\text{droplet } 1/2} = 33.6$ min), to bacterially expressed GFP (Nguyen Ba et al., 2009).

b, Experimental procedure. We expressed GFP with the predicted NLS signals from Yy1, Gtf2h1, and Gtf3a, and performed nuclear import assays. There, we added the newly expressed construct at the same concentration as the standard SV40-NLS-mCherry as a positive control into *Xenopus* egg extracts doped with sperm DNA. A time-series sample is collected and fixed at a 5 min interval; the collect time points are imaged with confocal microscopy.

c, We monitored the nuclear import kinetic of NLS-GFP and SV40-NLS-mCherry, and fit the data with a sigmoid to extract the time ($T_{1/2}$). We calculate the import time difference (ΔT) between mCherry- NLS and the NLS of interest to overcome extract variability. Markers represent the raw measurements, and the box plot shows the spread of measurement data for all the nuclei at each time point. Lines are sigmoid fits. From these experiments, we extract the median ΔT for each NLS construct.

d, A scatter plot summarizes the result from the import kinetics of GFP with transferred NLS and import kinetics of the corresponding protein. We observe a good correlation (Pearson correlation of 0.94, p-value = 0.006) between $T_{1/2}$ of NLS-GFP constructs measured in the bulk extract and the $AT_{\text{droplet } 1/2}$ of proteins measured in the droplet assay.

- 8) Another essential test would be to increase the concentration of importins in the cytoplasm. This may be technically ambitious in the intact oocytes, but for sure doable in the droplet assay. If importins are limiting, as postulated by the authors, there should be a level of importins at which the speed of import of different cargoes gets equalized.

We agree that this would be a great experiment, but unfortunately, it is not technically feasible: The native importin concentration is $\sim 2\mu\text{M}$ (Supplementary Table 4). The concentration of all importin substrates is $> 300\mu\text{M}$ (Supplementary Table 4). To perform this experiment, we would need to increase importin concentrations more than 100-fold above its physiological level $>300\mu\text{M}$. For comparison, the maximal solution concentration we can obtain when purifying importin **13** before crashing out is $30\mu\text{M}$, $\sim 10\times$ lower.

9) Fig. 5: Which of the nine proteins used in the droplet assay is shown in Fig. 5B? Fig. 5C shows only the order of appearance in the nucleus in both assays, not T1/2. Please blot the measured numbers to get a reliable correlation.

We thank Reviewer 1 for pointing out that the representation of the example protein Phf5a (illustrated in Fig. 5a,b) is not clear in Fig. 5c. We updated figure 5 to make this more clear.

Figure 5: The temporal order of nuclear import in cell-free droplets observed via imaging recapitulates the nuclear entry order observed in the embryo proteomic assay.

a, Left: Imaging of nuclear import in cell droplets. Xenopus egg extract doped with sperm DNA, which initiates the formation of nuclei, and GFP-tagged protein of interest

(here Phf5a) and mCherry-NLS were encapsulated in oil droplets with a microfluidic device.

Right: We monitored nuclear import kinetics via fluorescence microscopy.

b, We quantified the relative fluorescent signal intensity in the nucleus and cytoplasm and fit the data with a sigmoid to extract the time ($T_{\text{droplet}1/2}$) at which the relative intensity reaches half of its max value. To overcome extract variability, we calculate the import time difference ($\Delta T_{\text{droplet}1/2}$) between mCherry-NLS and the protein of interest. Markers represent the raw measurements. The different symbols represent different droplets; lines are sigmoid fits of corresponding droplets. From these experiments, we extract the median $\Delta T_{\text{droplet}1/2}$.

c, Scatter plot of the order in nuclear import time ($\Delta T_{\text{droplet}1/2}$) from the cell-free assay and the order in $T_{\text{embryo}1/2}$ for the nine TFs show strong agreement (Spearman correlation of 0.82, p-value = 0.007).

d, Imaging results show the concentration of early titrator nuclear protein (Yy1) is high at the early stage and decreases over time, followed by Gtf2e2 and Gtf2b.

e, In our import model, we predict that nuclear concentration of high-affinity proteins (blue) is high in the early nuclei and decreases with the continuous import of additional nuclear proteins and the increasing nuclear volume. Nuclear proteins with lower affinities (brown then orange) will reach their highest nuclear concentration at some later times and in a sequence corresponding to their interaction strengths to importin. The imaging results are consistent with the model. $[\text{nuclear } P_i]$ is the nuclear concentration of protein i ($i = \text{blue, brown, orange}$) that is being evaluated.

Furthermore, to address the reviewers' request to replot Fig. 5c in the absolute $\Delta T_{\text{droplet}1/2}$ value, we have added the panel below to Extended Data Fig. 9, panel d, which compares and measures the correlation between $\Delta T_{1/2}$ from the cell droplet assay and the embryonic assay.

Extended Data Figure 9d, Scatter plot of the nuclear import log time ($\log T_{\text{droplet1/2}}$) from the cell-free assay and the log import time $T_{\text{embryo1/2}}$ for the nine TFs show strong agreement (Pearson correlation of 0.77, p-value = 0.01).

Reviewer #2 (Remarks to the Author):

As early embryonic divisions create more and more nuclei, an increasing fraction of maternally deposited protein is imported into the nucleus. In this study, the authors use proteomic mass spec to monitor the nuclear proteins over time during early embryo development to determine whether some proteins are imported into the nucleus before others. After establishing that the import of nuclear proteins is ordered, the authors highlight how their measurements are consistent with what is already known. Finally, they pursue a mechanistic explanation for their measurements by assessing the importin binding affinities.

Overall, this is a very elegant use of proteomics. The data in this manuscript is presented clearly, and the results of the study are interesting and sensible. I recommend publication if the following points are addressed.

Major point:

- 1) From what I can tell, the authors only performed the primary experiment of this manuscript 1 time (though there are some columns in the supplement with the number "2" from non-overlapping timepoints). The authors must perform at least one independent biological replicate and then correlate the $T_{-1/2}$ s calculated from each experiment to demonstrate the degree of reproducibility. It would also be important to visualize where the individual proteins highlighted in the manuscript lie within the correlation plot. This replicate analysis may also help improve the correlation between the import affinity and $T_{-1/2}$.

Thanks for the suggestion. We would like to point out that in the original submission, the $T_{\text{embryo}1/2}$ is calculated using measurements from two independent biological replicates, though not all time points were perfectly matched. To address the Reviewer's concern, we have added three biological replicates of the primary experiment in this resubmission. The manuscript now contains five independent biological experiments, with a total of 18 time points, of nucleocytoplasmic partitioning as a function of developmental progression (each measured nucleus and cytoplasm) (Supplementary Table 2). We added a panel to Extended Data Figure 3a, (shown below) to demonstrate the level of reproducibility of the embryonic nuclear proteome experiments between two independent biological replicates. The Pearson correlation of $T_{\text{embryo}1/2}$ between these replicates are $R = 0.77$ (p-value $< 1E-325$).

Extended Data Figure 3a, Scatter plot of two biological replicates to determine $T_{1/2}$ of nuclear entry into embryonic nuclei. We observe a Pearson correlation of 0.81 (p-value $<1E-325$) with lower reproducibility at very late $T_{1/2}$ s.

To better visualize the subsets of individual proteins highlighted throughout the manuscript, each were isolated and shown in separate correlation plots (see below). The majority of these proteins are transcription factors that are low in abundance and often hard to detect in an MS single experiment. To achieve the proteome depth demonstrated in the paper, we combined data points from 2 (initial submission) or 3 (newly added measurements) replicates to extract a single $T_{embryo1/2}$ (we called the bulk replicate $T_{embryo1/2}$). Therefore, to best demonstrate the reproducibility of individual groups of highlight proteins, we compare data of experiments collected in the first submission (bulk replicate 1) versus experiments in this revision (bulk replicate 2). Overall, we achieve a good agreement of the highlighted proteins between the bulk replicates (the overall Pearson $R = 0.85$, p-value = $4e-29$ and the overall Spearman $R = 0.91$, p-value = $3e-38$)

Extended Data Figure 3b-i, Correlation plots for highlighted proteins discussed in the manuscript are separated into different panels, corresponding to the associated figures in the main text. Below are the pairs of the presented panel and the corresponding figure in the manuscript: panel (b) – Fig. 2b, panel (c) – Fig. 2e & Extended Data Fig. 3b, panel (d) – Fig. 3a, panel (e) – Fig. 3b & Extended Data Fig. 4, panel (f) - Fig. 3c, panel (g) – Fig. 5c, panel (h) – Extended Data Fig. 3a, panel (i) – Fig. 2d.

The correlation between the importin affinity and the updated $T_{embryo1/2}$ measurements, considering the additional biological replicates, remains unchanged (Fig. 4c). We believe this is likely because the main source of noise in the comparison comes from the importin affinity assay and because the additional measurements indeed suppress noise for proteins that were measured multiple times but at the same

time quantify new proteins (typically low abundant) that were only measured once or twice.

Minor points:

- 1) Are there predictions for the strength of a nuclear import sequence? Do proteins containing clear consensus NLS sequences among the first imported into the nucleus?

We thank Reviewer #2 for suggesting looking into the potential consensus sequence of the preferred nuclear-imported proteins from our measurements. To address the comment, we ran a MEME analysis on proteins with strong interaction to importin given a background of our negative set (proteins that are inert to the addition of RanQ69L molecule in our importin affinity assay). While the bioinformatic analysis provided a significant E-value of $\sim 5e-5$, it is unclear what the meaning of this sequence is. We would like to point out that the detection of NLS sequences is quite challenging on the sequence level. For instance, we found that bioinformatically predicted NLSs are nearly as likely to be present in the nucleus as in the cytoplasm. (Nguyen Ba *et al.*, 2009; Nguyen *et al.*, 2019). When analyzing this data further, we found that many bioinformatically predicted NLSs are, for example, either buried inside protein complexes, interact with RNA in the cytoplasm, or located inside mitochondria.

a, MEME analysis shows enrichment for only one motif with E-value of $5e-5$.
b, Previously, we observed that less than 60% of nuclear proteins in the oocyte are bioinformatically predicted to carry an NLS. Furthermore, these predicted NLS proteins are nearly as likely to be present in the nucleus or cytoplasm (red line) in the frog oocyte (Nguyen *et al.*, 2019; Wuhr *et al.*, 2015).

- 2) Do the binding affinities of the individual proteins highlighted in Figures 2 and 3 agree with the overall model. It would be nice to revisit those proteins in Figure 4.
 - We thank Reviewer #2 for suggesting adding connections between our embryonic time series observation and the measured importin affinity. However, we ran into a technical limit of the affinity pull-down assay, which cannot detect

low abundance proteins such as those mentioned in Fig. 2 and 3. Among the proteins listed in Fig. 2 and 3, we only observe a total of five individual proteins in the affinity assay. For these five proteins, the proposed trend between the $T_{\text{embryo1/2}}$ and the importin affinity proxy seems to hold (please see the following figure). However, we believe these are two few data points to revisit considering the spread of the entire scatter plot.

A few proteins and protein groups that we discussed in Fig. 2 and Fig. 3 are observed in the final data set of the importin assay. For the five detected proteins, we still observed a positive correlation between the nuclear import time and the importin affinity measurement.

- 3) Since it is such a striking observation, I encourage some speculation about the potential implications for the ordered import of the DNA repair proteins. Is it known whether certain DNA repair mechanisms are preferred over others during embryogenesis?

We again thank the Reviewer for this suggestion. We have modified the text in the manuscript to elevate the discussion on how different DNA repair mechanisms are preferred differentially by the early embryos as follows:

"Our observation of delayed sequential nuclear entry suggests that separating repair enzymes from DNA might contribute to the previously observed suppression of DNA repair during the rapid early cleavage cycles. (Fernandez-Diez et al., 2018; Hagmann et al., 1996; Kermi et al., 2019). Specifically, Hagmann et al. suggested that the DNA-end

joining (NHEJ) is dominant in the fertilized egg. However, with increasing amount of DNA with cell divisions homologous recombination (HR) becomes more prevalent."

- 4) The graphical abstract is confusing. It is unclear what the check marks are supposed to denote.

Thanks for pointing out the confusion in the graphical abstract. To clarify the meaning of the checks and the crosses, we have added a description of a transcript being "transcribed" next to a check and a transcript being "not transcribed" next to a cross. We hope that the graphical abstract is clear to understand now.

Reviewer #3 (Remarks to the Author):

The manuscript under review provides evidence for the attractive proposal that the relative affinities of proteins for their cognate importins is a major determinant in the timing of protein entry into embryonic nuclei. If correct, this is an important finding with major implications for our understanding of nucleocytoplasmic transport. Although the experimental evidence, with its emphasis on quantitative proteomics, is of high quality, there is a control that I believe is needed to test the validity of the lysate MS affinity assay - i.e., testing the assay with a set of NLS reporters of known affinity. I emphasize that this test is needed because the hypothesis rides on the integrity of the affinity assay.

We appreciate the Reviewer's constructive feedback. To address the Reviewer's comment, we performed an experiment to test our Ran-dependent importin assay using a set of control NLS peptide sequences (Hodel et al., 2001), whose importin affinities were previously measured using an orthogonal method - a fluorescence depolarization assay (Fanara et al., 2000). We repeated the importin assay used in our manuscript for the frog proteins using these peptides (Fig 4). When comparing our measured importin affinity proxy to the previously reported K_D values, we observed a strong Pearson correlation of 0.90 (i.e., $R^2 = 0.80$, p -value = 0.001). We added an Extended Data Fig. 7 to summarize the result described above.

Extended Data Figure 7. Validation of importin affinity assay (Fig 4) using NLS peptides with orthogonally measured K_D (Hodel *et al.*, 2001). We defined the importin affinity proxy as the free peptide concentration difference between a condition with RanQ69L and a condition without RanQ69L (importin affinity proxy = [free protein_{+RanQ69L}] - [free protein_{-RanQ69L}]). We observed a correlation of 0.90 ($R^2 = 0.80$, p -value = 0.001) between our measured importin affinity proxy with the $\log K_D$

measured by Hodel et al. (Hodel *et al.*, 2001). The dashed gray line is a linear fit. Error bars indicate standard error (n=4).

Reviewer #4 (Remarks to the Author):

The work by Nguyen et al sets up a methodology for systematically measuring time resolved nuclear import during embryonic development. They obtain quantitative temporal import measurements for close to 2000 nuclear proteins and convincingly illustrate that the import dynamics can to a significant extent explain the timing of transcriptional events during embryo development. Further they show that this sequential import can be rationalized in part by affinities of different nuclear proteins to importin. Finally, they set up an invitro assay to validate these findings.

This is beautiful work on many levels. The development of the improved nuclear purification protocol is impressive in terms of its performance and (Extended Data Figure 2). The temporal data is a very important resource that will be used by others in the field. Also, the biological findings and interpretation of the data are novel and exciting. I strongly recommend publication following minor revisions.

Thank you very much for these kind words. We greatly appreciate them!

Minor comments:

- 1) For the NPC data: currently, there is an extended figure comparing the basket proteins to other NPCs. As NPC is composed of several subcomplexes it would be interesting to illustrate and compare how different NPCs subcomplexes align or differ in their import dynamics.

The Reviewer is correct that the NPC is composed of several subcomplexes. Conventional classification of the NPC subcomplexes based on approximate localization across the nuclear envelope includes (1) a membrane-embedded core scaffold, (2) a central transport channel, (3) a cytoplasmic ring, and (4) a nuclear ring/basket (Hoelz et al., 2011; Raices and D'Angelo, 2012; Strambio-De-Castillia et al., 2010). When plotting the nuclear import time for these four subcomplexes, it seems like the order of entry minimally correlates with the relative position of the subcomplexes to the nucleoplasm or cytoplasm, e.g., the cytoplasmic filament enters at the lowest rate. In contrast, the nuclear basket is at the highest. However, the data are not statistically strong enough to infer a difference in the import rates between the core scaffold vs. central channels vs. cytoplasmic filament. This is why the main text only discusses the difference in the nuclear titration rate between the nuclear basket and the rest. We added the panel below to help visualize the discussion above.

The four main NPC subcomplexes nuclear entry time in early embryos

2) The manuscript mentions that often proteins part of the same complex are co-imported. This would be nice to represent more systematically eg by using the Corum annotation and checking the significance of co-import of complexes vs random assignment (for similar type of analysis for protein turnover you could look up this paper Mathieson et al Nat Comm 2018).

- Thanks, for the suggestion. We conducted a systematic analysis to test whether the chance for proteins from complexes is significantly higher than random assignment as described by (Mathieson et al., 2018) and show that complexes tend to co-enter the embryonic nuclei in a highly statistically significant fashion (p-value $8E-26$).
- First, based on the core complex data from the CORUM database we calculated the standard deviation of the $\log T_{\text{embryo}1/2}$ of all the components within a complex. We compare this to the equivalent analysis with a randomly shuffled protein complex list and perform Wilcoxon-rank test comparing the two datasets.

Extended Data Figure 4d, Protein complex subunits tend to co-import into the nucleus. The standard deviation of nuclear entry time of each complex measured in the embryonic assay (in blue) compared to those of complexes with shuffled subunits (in orange)(Mathieson *et al.*, 2018). Wilcoxon-rank test shows a highly significant distinction between the two distributions (p-value $\sim 8e-26$).

3) I would also appreciate some discussion on proteins that are getting imported but do not have nuclear annotation.

- Thanks for the suggestion. We have modified the main text to include the following text in the discussion:

"We observed several examples of proteins that were not nuclear in the oocyte 31 but were imported into the embryonic nuclei. Among those were the nuclear pore complex and the origin of the replication complex (Extended Data Fig. 4a), 13-catenin (Ctnnb1), and the CPC complex (Extended Data Fig. 4c)."

Extended Data Figure 4c, The CPC complex was equidistributed between the nucleus and cytoplasm in the oocyte. However, the CPC components seem to be sequestered into the embryonic nuclei. The transcription factor 13-catenin (Ctnnb1) is entirely cytoplasmic in the oocyte. However, 13-catenin is among the early nuclear-imported proteins (Griffin et al., 2018; MacDonald et al., 2009).

Blitz, I.L., and Cho, K.W.Y. (2021). Control of zygotic genome activation in *Xenopus*. *Curr Top Dev Biol* 145, 167-204. 10.1016/bs.ctdb.2021.03.003.

Bohnsack, M.T., Stuken, T., Kuhn, C., Cordes, V.C., and Gorlich, D. (2006). A selective block of nuclear actin export stabilizes the giant nuclei of *Xenopus* oocytes. *Nature cell biology* 8, 257-263. 10.1038/ncb1357.

Brownlee, C., and Heald, R. (2019). Importin α Partitioning to the Plasma Membrane Regulates Intracellular Scaling. *Cell* 176, 805-815 e808. 10.1016/j.cell.2018.12.001.

Callanan, M., Kudo, N., Gout, S., Brocard, M., Yoshida, M., Dimitrov, S., and Khochbin, S. (2000). Developmentally regulated activity of CRM1/XPO1 during early *Xenopus* embryogenesis. *J Cell Sci* 113 (Pt 3), 451-459. 10.1242/jcs.113.3.451.

Clark, T.G., and Merriam, R.W. (1977). Diffusible and bound actin nuclei of *Xenopus laevis* oocytes. *Cell* 12, 883-891. 10.1016/0092-8674(77)90152-0.

Fanara, P., Hodel, M.R., Corbett, A.H., and Hodel, A.E. (2000). Quantitative analysis of nuclear localization signal (NLS)-importin α interaction through fluorescence depolarization. Evidence for auto-inhibitory regulation of NLS binding. *J Biol Chem* 275, 21218-21223. 10.1074/jbc.M002217200.

Fernandez-Diez, C., Gonzalez-Rojo, S., Lombo, M., and Herraez, M.P. (2018). Tolerance to paternal genotoxic damage promotes survival during embryo development in zebrafish (*Danio rerio*). *Biol Open* 7. 10.1242/bio.030130.

Griffin, J.N., Del Viso, F., Duncan, A.R., Robson, A., Hwang, W., Kulkarni, S., Liu, K.J., and Khokha, M.K. (2018). RAPGEF5 Regulates Nuclear Translocation of 13-Catenin. *Dev Cell* 44, 248-260 e244. 10.1016/j.devcel.2017.12.001.

Guse, A., Fuller, C.J., and Straight, A.F. (2012). A cell-free system for functional centromere and kinetochore assembly. *Nat Protoc* 7, 1847-1869. 10.1038/nprot.2012.112.

Guttler, T., and Gorlich, D. (2011). Ran-dependent nuclear export mediators: a structural perspective. *The EMBO journal* 30, 3457-3474. 10.1038/emboj.2011.287.

Hagmann, M., Adlkofer, K., Pfeiffer, P., Bruggmann, R., Georgiev, O., Rungger, D., and Schaffner, W. (1996). Dramatic changes in the ratio of homologous recombination to nonhomologous DNA-end joining in oocytes and early embryos of *Xenopus laevis*. *Biol Chem Hoppe Seyler* 377, 239-250. 10.1515/bchm3.1996.377.4.239.

Hodel, M.R., Corbett, A.H., and Hodel, A.E. (2001). Dissection of a nuclear localization signal. *J Biol Chem* 276, 1317-1325. 10.1074/jbc.M008522200.

Hoelz, A., Debler, E.W., and Blobel, G. (2011). The structure of the nuclear pore complex. *Annu Rev Biochem* 80, 613-643. 10.1146/annurev-biochem-060109-151030.

Kermi, C., Aze, A., and Maiorano, D. (2019). Preserving Genome Integrity During the Early Embryonic DNA Replication Cycles. *Genes (Basel)* 10. 10.3390/genes10050398.

Kutay, U., Bischoff, F.R., Kostka, S., Kraft, R., and Gorlich, D. (1997). Export of importin α from the nucleus is mediated by a specific nuclear transport factor. *Cell* 90, 1061-1071. 10.1016/S0092-8674(00)80372-4.

Levy, D.L., and Heald, R. (2010). Nuclear size is regulated by importin α and Ntf2 in *Xenopus*. *Cell* 143, 288-298. 10.1016/j.cell.2010.09.012.

MacDonald, B.T., Tamai, K., and He, X. (2009). Wnt/ β -catenin signaling: components, mechanisms, and diseases. *Dev Cell* 17, 9-26. 10.1016/j.devcel.2009.06.016.

Mackmull, M.T., Klaus, B., Heinze, I., Chokkalingam, M., Beyer, A., Russell, R.B., Ori, A., and Beck, M. (2017). Landscape of nuclear transport receptor cargo specificity. *Mol Syst Biol* 13, 962. 10.15252/msb.20177608.

Mathieson, T., Franken, H., Kosinski, J., Kurzawa, N., Zinn, N., Sweetman, G., Poeckel, D., Ratnu, V.S., Schramm, M., Becher, I., et al. (2018). Systematic analysis of protein turnover in primary cells. *Nat Commun* 9, 689. 10.1038/s41467-018-03106-1.

Nguyen Ba, A.N., Pogoutse, A., Provart, N., and Moses, A.M. (2009). NLStradamus: a simple Hidden Markov Model for nuclear localization signal prediction. *BMC Bioinformatics* 10, 202. 10.1186/1471-2105-10-202.

Nguyen, T., Pappireddi, N., and Wuhr, M. (2019). Proteomics of nucleocytoplasmic partitioning. *Curr Opin Chem Biol* 48, 55-63. 10.1016/j.cbpa.2018.10.027.

Raices, M., and D'Angelo, M.A. (2012). Nuclear pore complex composition: a new regulator of tissue-specific and developmental functions. *Nat Rev Mol Cell Biol* 13, 687-699. 10.1038/nrm3461.

Strambio-De-Castillia, C., Niepel, M., and Rout, M.P. (2010). The nuclear pore complex: bridging nuclear transport and gene regulation. *Nat Rev Mol Cell Biol* 11, 490-501. 10.1038/nrm2928.

Stuven, T., Hartmann, E., and Gorlich, D. (2003). Exportin 6: a novel nuclear export receptor that is specific for profilin.actin complexes. *EMBO J* 22, 5928-5940. 10.1093/emboj/cdg565.

Wilbur, J.D., and Heald, R. (2013). Mitotic spindle scaling during *Xenopus* development by kif2a and importin α . *Elife* 2, e00290. 10.7554/eLife.00290 00290 [pii].

Wuhr, M., Guttler, T., Peshkin, L., McAlister, G.C., Sonnett, M., Ishihara, K., Groen, A.C., Presler, M., Erickson, B.K., Mitchison, T.J., et al. (2015). The Nuclear Proteome of a Vertebrate. *Curr Biol* 25, 2663-2671. 10.1016/j.cub.2015.08.047.

REVIEWER COMMENTS

Reviewer #1 (Remarks to the Author):

The authors have answered some of my comments but there are still major issues to be solved.

1) I acknowledge that the authors provide an elegant proteome-wide analysis of nuclear import kinetics in early frog development. They confirm that the nuclear entry of transcriptionally active proteins is correlated to the time point they are needed during differentiation of an embryo. Most researchers would have made time-dependent differential translational or post-translational modifications of each protein responsible for this phenomenon. The authors, however, postulate that mainly the affinity of the nuclear proteins to the importin- α 1-importin- β 1 complex explains this phenomenon. Their main evidence is the >40% correlation between this affinity and the time point of nuclear entry. I think this evidence is not sufficient to support the conclusions of the authors. Fig. 4C presenting this correlation shows that a very high number of proteins enters the nucleus very late ($T_{1/2} > 40$ hrs). How many are there? Do they enter at all? The same proteins are mainly in the low-importin-binder group. Is it possible that these are not at all nuclear proteins at least not in embryos and that they have been wrongly annotated? I think these proteins drive the main part of the correlation. If you would take them out there may be hardly any correlation left.

2) The authors have added novel data (Ext. Fig. 6) on the abundance of importins in the embryo, but strangely they exclude the α -importins (KPNA2), which they used to formulate their hypothesis, from the figures showing absolute abundance (6a) and fraction in nucleus (6b), they only appear in relative abundance (6c). All other importins shown (except KPNA1) have not been used in their affinity assay and may even be omitted. Please provide the data on KPNA2.

3) KPNA2 is also ignored in Fig. 4a, nuclear proteins bind first with their NLS to α importins and then the complex binds importin β . Thus, the assay depends on the affinity of each protein's NLS to KPNA2.

4) The DNA-affinity assay based on plasmid DNA is meaningless and can be deleted, since it does not explain the speed of nuclear entry. This is not surprising since nearly all nuclear proteins (maybe except histones) bind with high affinities to special sequences or to other proteins and the basic affinity to DNA is not relevant.

5) In order to test the main hypothesis of the paper, which is still based on a (weak) correlation, I suggested to perform some experiments. One of them was performed by the authors and shows that at least the NLS of three proteins behave like the proteins themselves in the droplet assay. The strong correlation shown in Ext. Fig. 10d is the first supporting experimental evidence of the authors' hypothesis. Still, it should have been performed with more than 3 NLS, for example also with the ones listed in the new Ext. Fig. 7, which would link the importin-affinity with the droplet assay. However the authors did not interfere with the importin level in the droplet assay, an experiment which I think is absolutely essential to prove that this level is crucial for the nuclear import kinetics of different proteins. They argue that they can't increase the concentration of the importins more than 100fold to reach the level of the substrates. I think any increase in importin concentrations (for example 10fold) should change the kinetics of the nuclear import and reduce the difference in import speed between high and medium affinity binders. The authors could predict the effect of such an increase using their mathematical model of the nuclear import kinetics. In the binding assay they use 10 μ M importins, such a concentration or even higher should therefore be possible in the droplet assay.

Reviewer #2 (Remarks to the Author):

The authors have addressed all my concerns in this extremely thorough and comprehensive revision.

Reviewer #3 (Remarks to the Author):

The authors have provided the control that I believed the paper needed, nicely demonstrating the

veracity of their assay. I am now confirmed in my belief that this is a absolutely superb paper - one which is likely become a classic in the field.

(My only recommendation would be to include Extended Data Fig 7 in the main body of the paper.)

REVIEWER COMMENTS

Reviewer #1 (Remarks to the Author):

The authors have answered some of my comments but there are still major issues to be solved.

1) I acknowledge that the authors provide an elegant proteome-wide analysis of nuclear import kinetics in early frog development. They confirm that the nuclear entry of transcriptionally active proteins is correlated to the time point they are needed during differentiation of an embryo. Most researchers would have made time-dependent differential translational or post-translational modifications of each protein responsible for this phenomenon. The authors, however, postulate that mainly the affinity of the nuclear proteins to the importin- α 1-importin- β 1 complex explains this phenomenon. Their main evidence is the >40% correlation between this affinity and the time point of nuclear entry. I think this evidence is not sufficient to support the conclusions of the authors. Fig. 4C presenting this correlation shows that a very high number of proteins enters the nucleus very late ($T_{1/2} > 40$ hrs). How many are there? Do they enter at all? The same proteins are mainly in the low-importin-binder group. Is it possible that these are not at all nuclear proteins at least not in embryos and that they have been wrongly annotated? I think these proteins drive the main part of the correlation. If you would take them out there may be hardly any correlation left.

Though we fully acknowledge that our reductionist model does not fully capture all the nuclear import regulations in the early embryo, our data are entirely consistent with a significant role of the proposed differential importin affinity model. This claim is supported by several results included in the manuscript:

1) We show that the total protein abundance for the vast majority of nuclear proteins are constant through the pre-MBT developmental period (Extended Data Figure 1b). This eliminates the production of new protein as the primary driver of changes in nuclear concentration.

2) We demonstrate that affinity to importin correlates with nuclear entry time (Figure 4c).

3) We demonstrate that our affinity assay can recapitulate orthogonally measured affinities to importin (Extended Data Figure 7).

4) We demonstrate that the order of entry observed in the developing embryo can be recapitulated in a single cell cycle (Figure 5). This indicates that developmentally timed changes in protein post-translational modifications on the timescales of multiple cell cycles are not the dominant drivers of nuclear import ordering.

Nevertheless, we clearly acknowledge in the manuscript the importance of other factors in regulating nuclear import beyond simply importin affinities:

“By necessity, our model ignores various layers of nuclear import regulation, which are most likely crucial for embryonic development, including substrate interactions with the other importins, exportins, and importin α ^{54,55}, changes of expression levels of all proteins involved, and post-translational modifications like phosphorylation of substrates or importins and exportins^{56,57}”

“Obviously, regulation of nuclear transport in the embryo must be much more complicated than implied by our simplistic assay and model.”

In addition, to address the reviewer's concern regarding the classification of nuclear proteins in the embryo, we have now tested how the proxy for importin affinity correlates with nuclear entry

time for various reasonable classifications of “nuclear proteins” and observed similar results (see figure below).

We agree with the reviewer that we do not have good access to a database for nuclear proteins in later-stage embryos, particularly considering the many different cell types they comprise. Nuclear protein databases will inherently be biased based on what cell types were used to generate them. To minimize this effect, we updated Fig. 4C with NLS-proteins (see below), which classification is only dependent on the protein sequence and agnostic to underlying cell types.

Figure 4: The affinity of proteins to importin contributes significantly to their ordering of nuclear entry in early development.

a, Estimation of proteome-wide affinity to importin $\alpha\beta$. We quantified changes of protein abundance associated with importin beads among conditions with varying amounts of RanQ69L. Abundance of known importin $\alpha\beta$ substrates⁵⁴, including histones, decreases with increasing RanQ69L concentration. Large dots represent the median protein fraction of a protein subgroup at each RanQ69L concentration, while small dots represent measurements for individual proteins. We applied a linear fit for each protein with a fixed y-intercept and used the slope to proxy for a protein's affinity to importin.

b, Scatter plot of triplicate affinity proxy measurements from experiments outlined in (a). We integrated these measurements to one dimension using cross-validated canonical correlation analysis⁵¹.

c, Importin affinity can explain a significant fraction of the timing of nuclear entry in early development. The scatter plot shows $T_{\text{embryo}1/2}$ versus importin $\alpha\beta$ affinity proxy. The observed Pearson correlation suggests that importin affinities can explain > 46% of the variance of the timing of nuclear entry or NLS containing proteins in early embryonic development.

d, Schematic of our proposed model in which the differential affinity of proteins to importin controls the timing of genomic access in embryonic development. A high-affinity protein titrates into the nucleus faster than a low-affinity protein, resulting in the corresponding DNA access of proteins. For proteins associated with transcription, this determines when certain transcriptional products appear.

e, Simulation of the model proposed in (d). We model competitive binding of substrates with varying affinity to a limiting number of importin. The proposed model provides a simple explanation for the timing of protein access to the embryonic genome in early development.

2) The authors have added novel data (Ext. Fig. 6) on the abundance of importins in the embryo, but strangely they exclude the alpha-importins (KPNAs), which they used to formulate their hypothesis, from the figures showing absolute abundance (6a) and fraction in nucleus (6b), they only appear in relative abundance (6c). All other importins shown (except KPNB1) have not been used in their affinity assay and may even be omitted. Please provide the data on KPNAs.

The data for absolute protein abundances for all measured proteins was available in Suppl Table 4 in the revised manuscript, but importin α s were not explicitly mentioned in a figure. To reconcile this, we have now included the absolute abundance measurements of the detected importin α s in the Extended Data Figure 6.

Extended Data Figure 6. The expression levels and subcellular localization of nuclear transport receptors observed in the frog embryos and their dynamic changes in early development.

a, The absolute abundance of transport receptors in the frog egg. Left: importin 13-like transport receptors, right importin α s (Supplementary Table S4⁴).

b, Relative protein abundance (left) and the subcellular localization (right) of the importin 13 family importins as a function of developmental progression. We observed that the expression

levels of importins stay constant throughout development and that the proteins locate preferentially in the cytoplasm throughout early development. The preferential cytoplasmic partition of importins was similarly observed in the frog oocyte⁴

, Relative protein abundances of importin α 's in early frog embryos show that their levels remain approximately constant.

a, Relative protein abundance changes of exportins and biportins as a function of developmental progression. Most exportins and biportins remain approximately constant. The exception is exportin 6 (Xpo6), which has actin as the primary substrate. Xpo6 is absent in *Xenopus* oocytes, which results in nuclear actin localization that supports the physical integrity of the large oocyte nucleus.^{14,15} Upon fertilization, Xpo6 expression level increases, and actin is excluded from embryonic nuclei.

3) KPNA2 is also ignored in Fig. 4a, nuclear proteins bind first with their NLS to alpha importins and then the complex binds importin beta. Thus, the assay depends on the affinity of each protein's NLS to KPNA2.

We thank the reviewer for making us aware of this unintentional omission. We have increased the detail of representation in the cartoon. The cartoon now also includes importin α :

4) The DNA-affinity assay based on plasmid DNA is meaningless and can be deleted, since it does not explain the speed of nuclear entry. This is not surprising since nearly all nuclear proteins (maybe except histones) bind with high affinities to special sequences or to other proteins and the basic affinity to DNA is not relevant.

We respectfully disagree with the reviewer on this point. Even though for technical reasons we were only able to use plasmid DNA to assay the proteins' affinity, the information adds predictive power when we use cross-validated projections (Extendend Data Figure 8). We note that we were explicitly clear about how the experiment was set up, the rationale for the chosen approach, and its inherent limitations. For these reasons, we argue that this supplementary

figure should be included and that readers should be allowed to draw their own conclusions regarding this data. However, we acknowledge that though this data suggests that DNA affinity seems to contribute to the timing of nuclear entry, we currently have no way to integrate this knowledge into a meaningful model. We believe this further justifies our reductionistic approach for the modeling.

5) In order to test the main hypothesis of the paper, which is still based on a (weak) correlation, I suggested to perform some experiments. One of them was performed by the authors and shows that at least the NLS of three proteins behave like the proteins themselves in the droplet assay. The strong correlation shown in Ext. Fig. 10d is the first supporting experimental evidence of the authors' hypothesis. Still, it should have been performed with more than 3 NLS, for example also with the ones listed in the new Ext. Fig. 7, which would link the importin-affinity with the droplet assay. However the authors did not interfere with the importin level in the droplet assay, an experiment which I think is absolutely essential to prove that this level is crucial for the nuclear import kinetics of different proteins. They argue that they can't increase the concentration of the importins more than 100fold to reach the level of the substrates. I think any increase in importin concentrations (for example 10fold) should change the kinetics of the nuclear import and reduce the difference in import speed between high and medium affinity binders. The authors could predict the effect of such an increase using their mathematical model of the nuclear import kinetics. In the binding assay they use 10 μM importins, such a concentration or even higher should therefore be possible in the droplet assay.

Though we indeed believe that the results of the suggested experiment of increasing importin concentration might prove interesting, as mentioned in our previous response, the experiment is not technically feasible at a level that would allow us to obtain interpretable data.

As the reviewer suggested, we have used mathematical modeling to demonstrate more clearly why we think the experiment is unlikely to produce meaningful results. Because the cumulative concentration of substrates binding to importins are much larger than the concentration of importin ($\sim 320\mu\text{M}$ for all NLS containing proteins vs. $\sim 1.5\mu\text{M}$ for importin β), we used the Langmuir model to simulate competitive binding. In the Langmuir model, the absolute concentration of importin $[I_0]$ cancels out (Eq. 7 from the supplement). The fractions of bound substrates θ_i will only depend on the dissociation constants $K_{i,0}$. We have introduced a modified θ_i^* (Eq. 7') to test the effect of enhanced Importin concentration $[I_0]$.

$$\theta_i = \frac{[I_0] P_{i,0}}{[I_0] + \sum_j \frac{[I_0] P_{j,0}}{K_{j,0}}} \quad 7$$

$$\theta_i^* = \min \left(\theta_i, \frac{[I_0] P_{i,0}}{[I_0] + [I_0]} \right) \quad 7'$$

$$\frac{dP_{nuc,i}}{dt} = e_i F_{total} = e_i \frac{dV_{nuc}}{dt} \sum_{k=1}^n [P_k]$$

8'

Now, the fraction of bound substrates e_i is limited by the maximum number of possible binders and for simplicity binders are depleted as fast as possible (which is an overestimation of the expected effect). e_i is normalized to 1 by rescaling the fractions of non-depleted binders. We then compared the predictions for varying $[I_o]$ (Figure below), keeping the total flux across the nuclear envelope F_{total} constant for comparability (Eq. 8'). Increasing $[I_o]$ from 1.5uM (solid lines) to 10uM (dotted lines), the model predicts a slight speed-up of the import of high affinity substrates ($K_D=2nM$) and only subtle changes for lower affinities ($K_D=20nM$ and 200nM). Based on the experimental variability in our nuclear import assay (Fig. 5d), we do not expect these effects to be distinguishable experimentally, even if we would invest the very significant time and effort to perform these experiments.

We acknowledge that it is an open question in our system as to how F_{mr_j} would change with increasing importin concentrations. Nuclear import is a complex interplay of many factors, and it is unclear what limits the total import rate. For example, it could be importins, Ran gradient generation system, or the number of available nuclear pore complexes (PMID: 18048681). That the number of nuclear pores being limited is somewhat supported by our findings that, in the embryo and extract system, total nuclear flux first increases with nuclear surface increase despite importin concentrations being constant (Extended Data Figure 9 a, c). Regardless, any increase in the rate of total nuclear flux would only result in contracting the x-axis above but would not change the relative order of import. We very much believe that manipulating importin concentrations and observing how nuclear transport rates change is an exciting though challenging experiment. Nevertheless, the questions addressed go clearly beyond the scope of this paper.

We have acquired some preliminary data testing the limit at which extract could be diluted for the nuclear import assays. The preliminary results suggest that dilution is impossible beyond 30% without severely disturbing nuclear import. We believe this a very optimistic preliminary number that is very likely to be lower for kinetic importin assays. As a rule of thumb, dilution beyond 10% often disrupts cell biological processes in *Xenopus* egg extract. Assuming 30% dilution is indeed workable, with experimentally achievable stock solutions for the importins we therefore can only achieve importin concentrations in the extract of ~10 μ M. This is about an order of magnitude lower concentration than we would need to lose the expected rank ordering of different nuclear import substrates.

We chose the proteins from Figure 5 for Extended Data Fig 10 because they had a bioinformatically identifiable NLS. The remaining proteins either showed no bioinformatically identifiable NLS or had multiple NLSs. In both cases, we could not transfer the sequence that conveys the import signal to a fluorescent protein. Regardless, the p-value of the measured proteins in Extended Data Figure 7 led to a highly significant p-value of 0.006, very strongly suggesting that indeed the entry kinetics of the tested proteins can be transferred to GFP via the predicted NLS. We do not believe acquiring more data points would alter our conclusions based on the already convincing statistics.

Overall, we are delighted that reviewer #1 seems to agree with two of the main findings of our paper, namely that (i) we demonstrate nuclear proteins enter embryonic nuclei at different times in early development and (ii) that the entry of nuclear proteins correlates with downstream function, suggesting that this is likely a timing mechanism used by the developing embryo. We have put forward a simplified model that can be used to explain observed differences in nuclear entry timing. As mentioned above, the paper provides multipronged evidence for this model though we fully acknowledge that it is a simplification of a complex phenomenon. Furthermore, our data demonstrate that total embryo-wide nuclear protein levels are typically constant in early development. Therefore, nuclear composition changes are primarily due to changes in protein partitioning, not production. We show the correlation of nuclear entry time with importin affinities and validate the affinity assay for orthogonally measured peptide affinities. We show that differential timing can be recapitulated within a single cell cycle. Via imaging, we observe entry kinetics that are similar to the predicted kinetics by the model with a defined maximum for early entry proteins. In summary, we hope that reviewer #1 agrees with the other three reviewers and us that this paper, despite the many open questions it raises, sufficiently advances the field to merit publication.

Reviewer #2 (Remarks to the Author):

The authors have addressed all my concerns in this extremely thorough and comprehensive revision.

We thank reviewer #2 for the kind words and greatly appreciate them.

Reviewer #3 (Remarks to the Author):

The authors have provided the control that I believed the paper needed, nicely demonstrating the veracity of their assay. I am now confirmed in my belief that this is an absolutely superb paper - one which is likely to become a classic in the field.

(My only recommendation would be to include Extended Data Fig 7 in the main body of the paper.)

We are thrilled by the encouragement from reviewer #3 and appreciate this suggestion. However, after some consideration, we believe Extended Data Fig. 7 is too technical for most readers, and we believe it is better to keep it in the supplement.